# Aberrant association of chromatin with nuclear periphery induced by Rif1 leads to mitotic defect

Yutaka Kanoh[1] , Masaru Ueno[2] , Motoshi Hayano[3], Satomi Kudo[1], Hisao Masai[1]

The architecture and nuclear location of chromosomes affect chromatin events. Rif1, a crucial regulator of replication timing, recognizes G-quadruplex and inhibits origin firing over the 50–100-kb segment in fission yeast, *Schizosaccharomyces pombe*, leading us to postulate that Rif1 may generate chromatin higher order structures inhibitory for initiation. However, the effects of Rif1 on chromatin localization in nuclei have not been known. We show here that Rif1 overexpression causes growth inhibition and eventually, cell death in fission yeast. Chromatin-binding activity of Rif1, but not recruitment of phosphatase PP1, is required for growth inhibition. Overexpression of a PP1-binding site mutant of Rif1 does not delay the S-phase, but still causes cell death, indicating that cell death is caused not by S-phase problems but by issues in other phases of the cell cycle, most likely the M-phase. Indeed, Rif1 overexpression generates cells with unequally segregated chromosomes. Rif1 overexpression relocates chromatin near nuclear periphery in a manner dependent on its chromatin-binding ability, and this correlates with growth inhibition. Thus, coordinated progression of S- and M-phases may require regulated Rif1-mediated chromatin association with the nuclear periphery.

## Introduction

Chromosomes are packaged and compartmentalized in the nuclei in an organized manner and their spatial arrangement and regional interactions regulate various chromosome transactions. Presence of chromosomes near the nuclear periphery is generally associated with inactive transcription or late replication (Masai et al, 2010; Lemaitre & Bickmore, 2015; Mahamid et al, 2016). The nuclear lamina, nuclear pore complexes, and epigenetic regulators have been implicated in regulating tethering of chromatin at the nuclear periphery (Arib & Akhtar, 2011; Ptak & Wozniak, 2016; See et al, 2020; Amiad-Pavlov et al, 2021; Laghmach et al, 2021; Smith et al, 2021). Rif1 was identified as a regulator of replication timing and was shown to suppress late origin firing. On the basis of the effect of a Rif1-

binding site mutant on origin firing patterns and biochemical functions of the Rif1 protein, we have proposed that Rif1 regulates replication timing partly through generating specific chromatin architecture in fission yeast, *Schizosaccharomyces pombe* (Kanoh et al, 2015; Kobayashi et al, 2019; Masai et al, 2019). In mammalian cells, Rif1 is localized near the nuclear periphery and was shown to regulate chromatin loop formation (Cornacchia et al, 2012; Yamazaki et al, 2012; Foti et al, 2016). However, how Rif1 regulates chromatin localization and its potential roles in regulation of cellular events have not been well known. We have tested this by examining the effects of overexpression of Rif1 on the cell cycle progression and chromatin states in fission yeast cells.

Rif1, originally isolated as a telomere-binding factor, is a multifunctional protein that regulates various aspects of chromosome dynamics, including DSB repair, DNA replication, recombination, transcription, and others (Cornacchia et al, 2012; Hayano et al, 2012; Yamazaki et al, 2012; Campolo et al, 2013; Di Virgilio et al, 2013; Zimmermann et al, 2013; Klein et al, 2021; Yoshizawa-Sugata et al, 2021). Fission yeast Rif1 binds to chromatin, most notably at the telomere, and telomeres are elongated in *rif1Δ* cells (Kanoh & Ishikawa, 2001; Kobayashi et al, 2019). Hayano et al reported that *rif1Δ* can restore the growth of *hsk1Δ* cells (Hayano et al, 2012), indicating that the loss of *rif1* can bypass the requirement for the fission yeast "Cdc7" kinase (Dbf4-dependent kinase; DDK), which is essential for replication initiation under normal growth conditions. Late-firing origins are extensively deregulated in *rif1Δ* cells, consistent with a role for Rif1 in suppressing late-firing origins. Similarly, mammalian Rif1 was also found to regulate the genome-wide replication timing (Cornacchia et al, 2012; Yamazaki et al, 2012; Klein et al, 2021). Rif1 interacts with PP1 phosphatase through its PP1-binding motifs, and the recruitment of the phosphatase by Rif1 counteracts the phosphorylation events that are essential for initiation of DNA replication, including the phosphorylation of Mcm, explaining the mechanism of Rif1-mediated inhibition of replication initiation (Dave et al, 2014; Hiraga et al, 2014; Mattarocci et al, 2014; Shyian et al, 2016).

In addition to its binding to telomeres, fission yeast Rif1 also binds to the arm segments of the chromosomes. Thirty-five strong Rif1-binding sites (Rif1bs) have been identified on fission yeast

[1]Department of Basic Medical Sciences, Tokyo Metropolitan Institute of Medical Science, Tokyo, Japan [2]Graduate School of Integrated Sciences for Life, Hiroshima University, Higashi-Hiroshima, Japan [3]Department of Neuropsychiatry, Keio University, Tokyo, Japan

Correspondence: masai-hs@igakuken.or.jp

chromosomes. These sequences contain multiple G-tracts and have the propensity to form G-quadruplex (G4) structures (Kanoh et al, 2015). Consistent with this, Rif1 specifically binds to G4-containing DNA in vitro, and mutations of G-tracts impaired both in vivo chromatin binding of Rif1 and in vitro interaction of Rif1 with the Rif1bs. Notably, loss of Rif1 binding at a single Rif1bs caused deregulation of late-firing origins in the 50–100-kb segment in its vicinity, consistent with the notion that Rif1 binding generates a chromosome compartment where origin firings are suppressed. In fission yeast, Rif1 was implicated also in the resolution of non-telomeric ultrafine anaphase bridges (Zaaijer et al, 2016).

In mammals, Rif1 is preferentially localized at the nuclear periphery in the Triton X-100– and DNase I–resistant compartments, where it regulates the length of chromatin loops (Yamazaki et al, 2012, 2013). In fission yeast, Rif1 is biochemically fractionated into Triton X-100– and DNase I–insoluble fractions (Kanoh et al, 2015). In budding yeast, Rif1 was shown to be palmitoylated, and the lipid modification–mediated membrane association plays important roles in DSB repair (Park et al, 2011; Fontana et al, 2019). However, it is unknown whether similar mechanisms operate for Rif1 from other species.

We hypothesized that Rif1 generates higher order chromatin architecture through its ability to tether chromatin loops and that this chromatin structure constitutes the replication inhibitory chromatin compartments that are deregulated during the mid-S phase. To gain more insight into the roles of Rif1 in regulation of the chromatin structure and cell cycle, we have analyzed the effect of Rif1 overexpression on the growth, cell cycle progression, and chromatin structure in fission yeast, *S. pombe*. The results indicate that regulated association of chromatin with the nuclear periphery may play a crucial role in proper S- and M-phase progression.

# Results

## Overexpression of Rif1 prevents cell growth

The *rif1* mutation was identified as a suppressor of the *hsk1-null* mutation (encoding Cdc7 kinase [DDK] homologue) in fission yeast, *S. pombe* (Hayano et al, 2012) and we previously showed that Rif1 suppressed origin firing over ~100-kb segments spanning its binding sites (Kanoh et al, 2015). During the course of our experiments, we cloned the *rif1$^+$* ORF into pREP41 and expressed Rif1 under the inducible nmt41 (no message in Thiamine 1) promoter (Fig 1A and B). Induction of the full-length Rif1 (1–1,400 aa) in a medium without thiamine strongly inhibited the growth of both *hsk1$^+$* and *hsk1-89* (temperature-sensitive stain) cells (Fig 1C and D). Various truncated deletion mutants of Rif1 were cloned and expressed from the inducible nmt41 promoter to examine growth in *hsk1$^+$* or *hsk1-89* cells (Fig 1B). The expression levels of these truncation mutants were examined by Western blotting, and the results indicate that all the mutants are uniformly expressed (Fig S1A). We measured the numbers of Rif1 molecules in yeast cells. As a standard, we first purified His$_6$–Rif1(93–1,400 aa)–Flag$_3$ because we found that deletion of the N-terminal 92 aa stabilized the protein (Moriyama et al unpublished data). The amount of Rif1–Flag$_3$

expressed at the endogenous locus and on a plasmid was assessed by Western blotting. The number of endogenous Rif1 molecules was estimated to be ~1,000, whereas that of plasmid-derived Rif1 was estimated to be 10,000 and 37,000 before and after induction, respectively (Fig S1B). A C-terminal 140–amino acid deletion (construct 1–1,260 aa) inhibited the growth of *hsk1$^+$* weakly and *hsk1-89* strongly (Fig 1C and D). Further truncations of the Rif1 C terminus (constructs 1–965 aa and 1–442 aa) resulted in complete loss of growth inhibition. We previously showed that truncation of the Rif1 C-terminal 140 amino acids (construct 1–1,260 aa) resulted in the loss of telomere length regulation (Kobayashi et al, 2019). Therefore, we conclude that the growth inhibition caused by overexpression of Rif1 does not depend on its function in telomere regulation. In contrast, deletion of the N-terminal 150 amino acids from Rif1 resulted in the loss of growth inhibition (Fig 1C and D). However, deletion of the N-terminal 80 amino acids did not affect the ability of Rif1 to inhibit the growth in the *hsk1$^+$* cells (Fig 1C), suggesting that the segment 81–150 aa is important for inhibition. Taken together, the results suggest that the segment 81–1,260 aa may be required and sufficient for inhibition. The N-terminal domain (88–1,023 aa) of fission yeast Rif1 is predicted to form a 3-D structure identical to that of the HEAT repeats by AlphaFold2 (Jumper et al, 2021). Thus, the HEAT domain 88–1,023 aa may play a major role for growth inhibition by Rif1, although it has not been experimentally tested if 88–1,023 aa is sufficient for inhibition.

## Growth inhibition by Rif1 overexpression does not involve recruitment of PP1

Rif1 recruits protein phosphatase 1 (PP1) through its PP1-binding motifs (Rif1$_{40-43}$ and Rif1$_{64-67}$), and the recruited PP1 counteracts the phosphorylation by Cdc7 kinase at origins (Dave et al, 2014; Hiraga et al, 2014; Mattarocci et al, 2014). This interaction of Rif1 with PP1 is crucial for replication inhibition by Rif1 at late origins. Therefore, we examined whether Rif1 overexpression–induced growth inhibition is caused by hyper-recruitment of PP1. Fission yeast cells have two PP1 genes: *dis2$^+$* and *sds21$^+$*. A single-disruption mutation of *dis2* or *sds21* is viable, but the double mutation is lethal (Kinoshita et al, 1990). *dis2-11* is a cold-sensitive mutant of *dis2$^+$*. We first examined whether growth inhibition caused by Rif1 overexpression depends on the PP1 genes. Overexpression of Rif1 in *dis2-11*, *dis2Δ*, and *sds21Δ* resulted in strong growth inhibition in all the tested strains on EMM media without thiamine (Fig 2A). The extent of inhibition in each PP1 mutant was as strong as that observed in the WT, suggesting that the recruited PP1 is not responsible for growth inhibition.

To further examine the involvement of PP1, we constructed a PP1-binding mutant of Rif1. We generated an alanine-substituted mutant of the two PP1-binding motifs (KVxF at aa 40–43 and SILK at aa 64–67) of Rif1 (Fig 2B) and confirmed, by immunoprecipitation, that the mutant Rif1 (PP1bsmut) did not bind to either Dis2 or Sds21 (Fig 2C; compare lanes 14 and 17, lanes 15 and 18). Growth inhibition caused by overexpression of PP1bsmut was comparable to or even slightly stronger than that caused by the WT Rif1 in both *rif1$^+$* and *rif1Δ* backgrounds (Fig 2D and E). This is a further support for the conclusion that growth inhibition by Rif1 overexpression does not depend on recruitment of PP1. These results are consistent with the

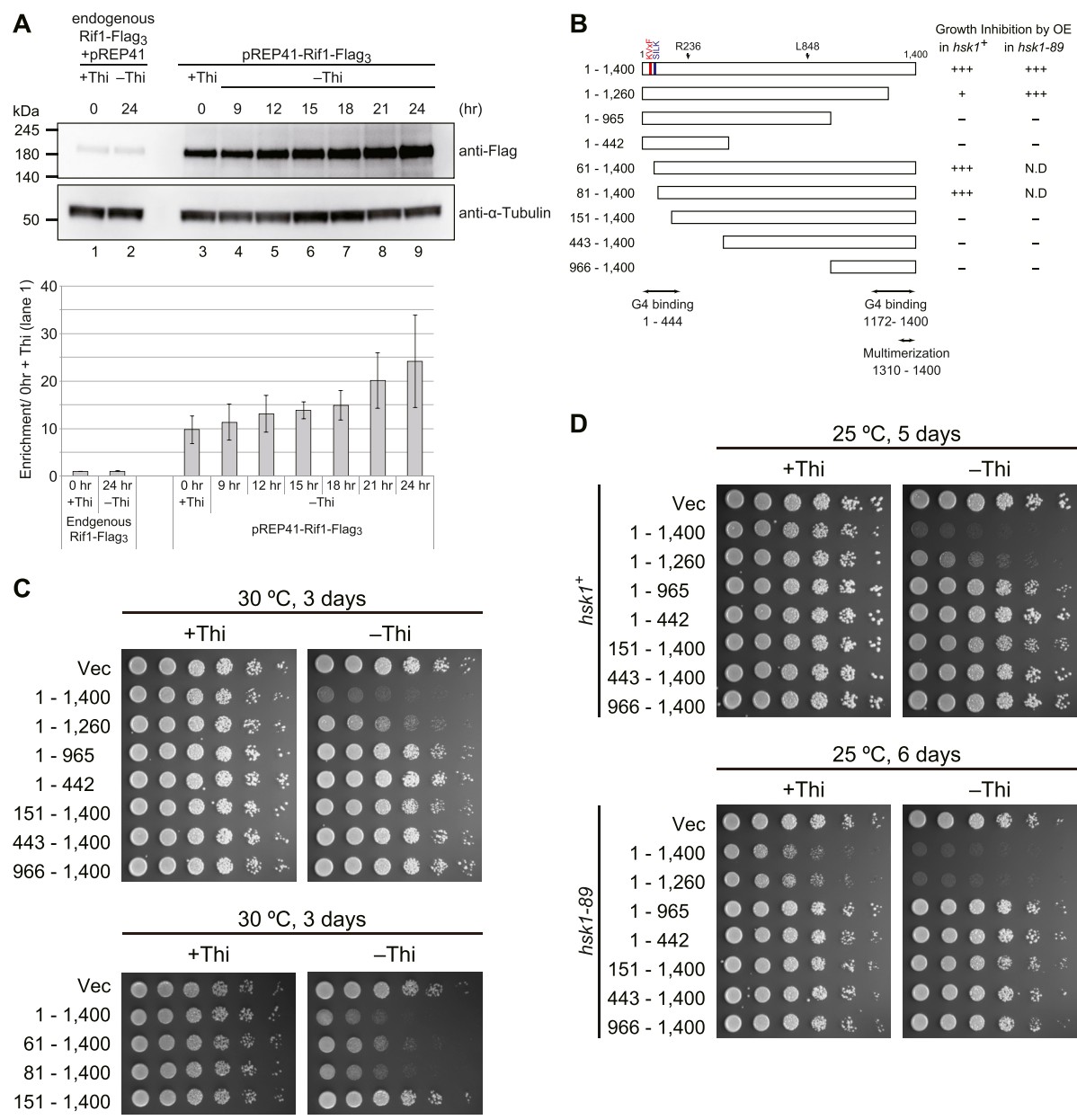

**Figure 1. Overexpression of Rif1 inhibits the growth of fission yeast cells.**
**(A)** Time course of overexpression of Rif1–Flag$_3$ protein expressed on pREP41 plasmid under the nmt41promoter after transfer to medium lacking thiamine (lanes 3–9) (KYP008 + pREP41–Rif1–Flag$_3$). Lanes 1 and 2 (HM511 + pREP41), Rif1–Flag$_3$ is expressed at the endogenous rif1 locus under its own promoter in the presence or absence of thiamine in the medium. Proteins were detected by the anti-Flag antibody. **(B)** Schematic drawing of deletion derivatives of Rif1 protein analyzed in this study (KYP008 + pREP41-Rif1 truncation series in Table 2). + indicates growth inhibition, whereas – indicates the absence of growth inhibition. The PP1-binding motifs (RVxF and SILK) are indicated in red and blue, respectively. Note that the motifs in Rif1 are slightly diverged from the above consensus sequences. The polypeptide segments capable of G4 binding and oligomerization are also indicated. **(C, D)** Effects of overexpression of the full length and truncated mutants of Rif1 were examined. Proteins were expressed on pREP41 in medium containing (+Thi) or lacking (–Thi) thiamine. Serially diluted (5× fold) cells were spotted and growth of the spotted cells was examined after incubation at the indicated temperature for the indicated time. Growth inhibition was observed with full-length (1–1,400) (KYP1805), 1–1,260 (KYP1853), 61–1,400 (KYP1806), or 81–1,400 (KYP1807) derivatives of Rif1.
Source data are available for this figure.

observation that the N-terminal truncation Rif1 (81–1,400aa), which lacks the PP1-binding sites, can inhibit the growth upon over-expression (Fig 1C). As shown in the later section, the PP1bs mutant of Rif1 loses the ability to inhibit DNA synthesis, indicating that growth inhibition by Rif1 is related to the events other than S phase.

**Growth inhibition by Rif1 overexpression does not depend on Taz1 or replication checkpoint**

We next asked whether the growth inhibition by overexpressed Rif1 is caused by counteracting the Hsk1-Dfp1/Him1 activity. Co-expression of

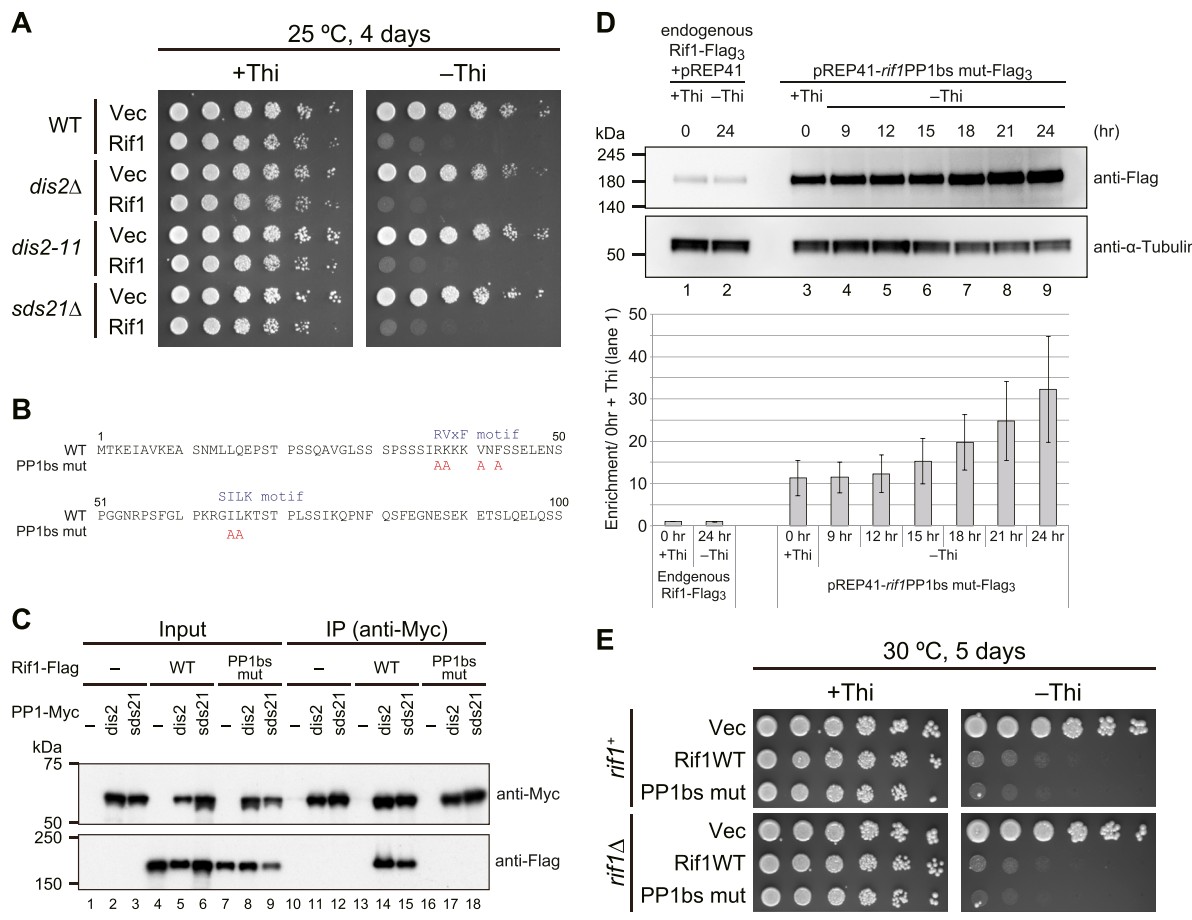

**Figure 2. PP1–Rif1 interaction is not required for the growth inhibition caused by Rif1 overexpression.**
**(A)** Spot tests of Rif1 overexpression in the PP1 mutants *dis2*Δ (KYP1762), *dis2-11* (KYP1760), or *sds21*Δ (KYP1764) cells were conducted as described in Figs 1C and D. Rif1 overexpression inhibited the growth of mutant cells similar to WT cells. **(B)** Mutations introduced at the PP1-binding sites (RVxF and SILK motif) of Rif1. **(C)** Using the extracts made from the cells expressing both Flag-tagged Rif1 and Myc-tagged PP1 (Dis2 or Sds21) (KYP1769, KYP1770, KYP1772, and KYP1773), PP1 were immunoprecipitated by anti-Myc antibody, and co-immunoprecipitated Rif1 was detected. The PP1bs mutant of Rif1 does not interact with either PP1. **(D)** Time course of overexpression of *rif1*PP1bsmut–Flag$_3$ protein expressed on pREP41 plasmid under the nmt41 promoter after transfer to a medium lacking thiamine (lanes 3–9) (KYP1839). Lanes 1 and 2 (KYP1827), *rif1*PP1bsmut–Flag$_3$ is expressed at the endogenous *rif1* locus under its own promoter in the presence or absence of thiamine in the medium. **(E)** Spot tests of the WT (KYP025, KYP015, and KYP1774) and *rif1*Δ (KYP1804, KYP1805, and KYP1839) cells overexpressing the WT or a PP1bs mutant. Overexpression of the PP1bs mutant Rif1 inhibited the growth of fission yeast cells in a manner similar to or slightly better than the WT Rif1.
Source data are available for this figure.

both Hsk1 and Dfp1/Him1 under the control of the nmt1 promoter itself caused growth inhibition in fission yeast cells, and growth was partially restored by *rif1* deletion. However, overexpression of Hsk1-Dfp1/Him1 did not improve the growth of Rif1-overexpressing cells and inhibited the growth more strongly (Fig S2A), excluding the possibility that inhibition of growth is due to the reduced Hsk1 kinase actions. Consistent with the results of the C-terminal deletion mutant (1–1,260 aa), which does not interact with Taz1 but still inhibits growth (Fig 1C and D), the mutation of *taz1*[+] known to be required for telomere-localization of Rif1 did not affect the growth inhibition by Rif1 overexpression (Fig S2B). Similarly, growth inhibition was observed in mutants of the replication checkpoint genes (Furuya & Carr, 2003), *rad3 tel1*, *rad3*, *chk1*, or *cds1* (Fig S2B). The extent of growth inhibition was not affected by *cdc25-22* or *wee1-50*, genes involved in mitosis (Rowley et al, 1992; Iino & Yamamoto, 1997; Kumar & Huberman, 2004) (Fig S2C). These results suggest that growth inhibition is not caused by

replication or mitotic checkpoint functions or deregulation of mitotic kinases.

### Effect of Rif1 overexpression on entry into the S-phase and replication checkpoint activation

To clarify the mechanisms of Rif1-mediated growth inhibition, we examined whether Rif1 overexpression inhibits S-phase initiation and progression. We synchronized the cell cycle by release from *nda3*-mediated M-phase arrest and analyzed the DNA content by FACS. In the WT cells, DNA synthesis was observed at 30 min (at 18.5 h in the FACS chart in Fig 3A) from the release, and continued until 19.5 h (90 min). In Pnmt41-Rif1, where Rif1 was overexpressed, DNA synthesis was delayed by 30 min (at 19.0 h in the FACS chart in Fig 3A) and was not completed even at 20 h (120 min), indicating that Rif1 overexpression retarded the initiation and elongation of DNA synthesis. In contrast, in Pnmt41-*rif1*PP1bs mut, DNA synthesis occurred

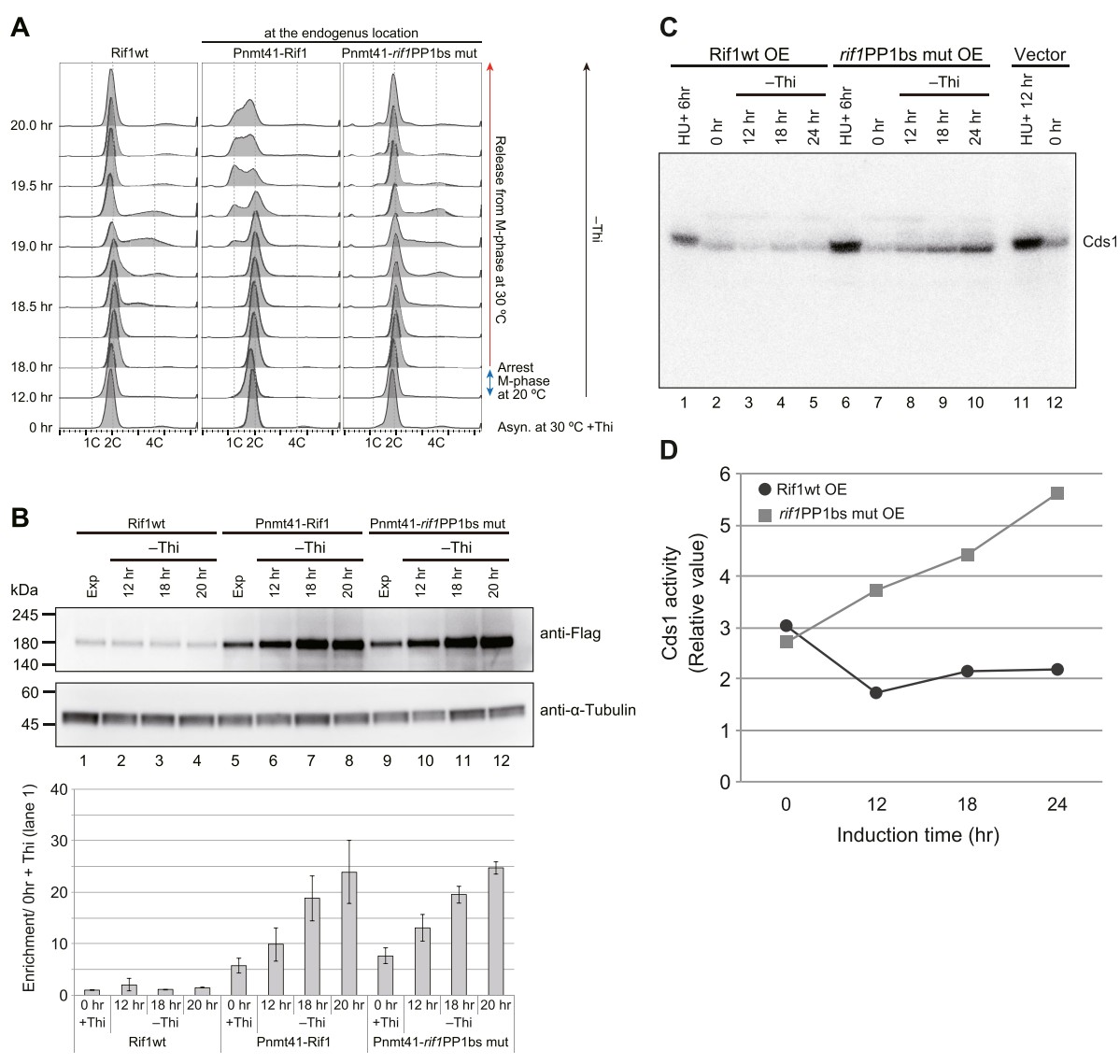

**Figure 3. Effect of overexpression of Rif1 protein on cell cycle progression and replication checkpoint activation.**
**(A)** The *nda3-KM311* cold-sensitive mutant cells with WT *rif1⁺* (KYP1268) or those expressing the WT Rif1 (Pnmt41–Rif1) (MS733) or PP1bs mutant Rif1 (Pnmt41–*rif1*PP1bsmut) (KYP1283) at the endogenous *rif1* locus under nmt41 promoter were arrested at the M-phase by incubation at 20°C for 6 h with concomitant depletion of thiamine. The cells were released into the cell cycle at 30°C. The cell cycle progression was monitored by flow cytometry. The cells with Pnmt41–*rif1*PP1bsmut entered the S-phase at 30 min (at 18.5 h in FACS chart) after release from M-phase arrest, similar to the *rif1⁺* cells, whereas those with Pnmt41-Rif1 entered the S-phase later (>60 min after release). **(A, B)** The level of Rif1 in the samples from (A) was examined by Western blotting. **(C)** The cells harboring Rif1 (wt or *rif1*PP1bsmut)-expressing plasmid or vector, as indicated, were starved for thiamine for the time indicated. The whole cell extracts were prepared and were run on SDS–PAGE containing MBP (Myelin Basic Protein) in the gel. In-gel kinase assays were conducted as described in the "Materials and Methods section." HU, treated with 2 mM HU for the time indicated as a positive control of Cds1 activation. **(C, D)** Quantification of the results in (C).
Source data are available for this figure.

with timing similar to the WT, indicating that the overexpression of the PP1 mutant Rif1 does not affect the S-phase (Fig 3A and B). This suggests that inhibition of the S-phase by overexpressed Rif1 is due to the hyper-recruitment of PP1, which would counteract the phosphorylation events mediated by Cdc7 or Cdk and inhibit initiation.

We next examined whether overexpression of Rif1 activates the replication checkpoint. We measured Cds1 kinase activity by in-gel kinase assay. Whereas Cds1 kinase activity decreased 12 h after induction and then slightly increased in cells overexpressing the WT Rif1, it continued to increase until 24 h after overexpression of the PP1bs mutant (Fig 3C and D). These results indicate that overexpression of the *rif1*PP1bs mutant activates the replication checkpoint, whereas the WT Rif1 does not, consistent with the above results that show growth inhibition is not caused by the replication checkpoint.

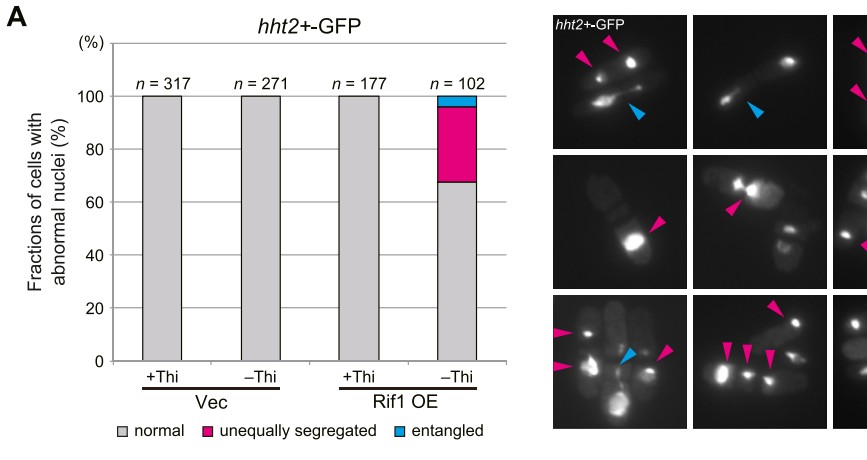

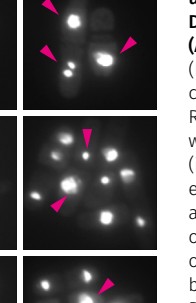

**Figure 4. Rif1 overexpression induces unequal chromosome segregation and DNA damages.**
**(A)** Chromosomes are visualized by *hht2+* (histone H3 h3.2)-GFP (right) and the chromosome segregation was assessed in Rif1-overexpressing yeast cells (KYP1776). Cells with unequally segregated chromosomes (indicated by mazenta arrowheads) or entangled chromosomes (indicated by blue arrowheads) increased at 24 h after Rif1 overexpression (left). **(B)** Rif1 was overexpressed in cells expressing Rad52–EGFP by depletion of thiamine for 24, 48, and 72 h. Rad52–EGFP foci in the cells were observed under fluorescent microscopy (KYP1777, KYP1778, KYP1860, and KYP1861). The numbers of Rad52 foci (representing DNA damages) were counted, and cells containing 0, 1, 2, or >3 foci were quantified. The extent of DNA damages increased with the duration of Rif1 overexpression.
Source data are available for this figure.

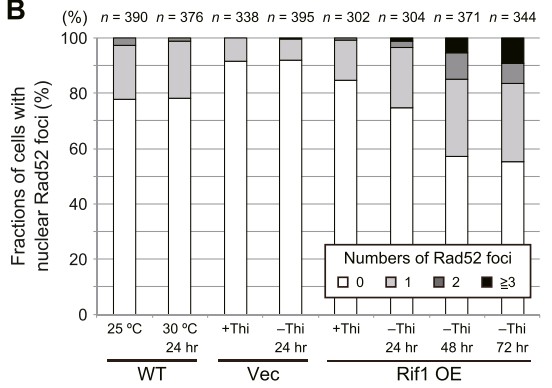

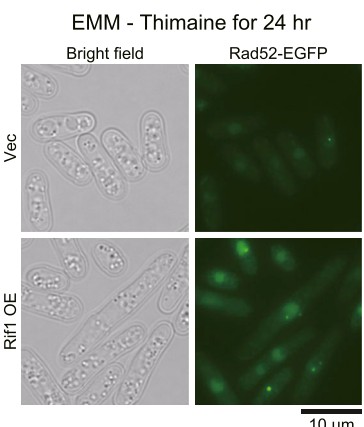

## Short spindles and abnormally segregated nuclei are accumulated in cells overexpressing Rif1

We next observed the morphological effects of Rif1 overexpression in cells expressing GFP-tagged histone H3 (*hht2+-GFP*) or GFP-tagged α-Tubulin (*GFP-α2tub*). Cells with aberrant morphology appeared in Rif1-overexpressing cells, indicative of the failure of chromosome segregation. At 22 h after induction, cells with abnormal nuclei accumulated and, notably, cells with unequally segregated nuclei reached ~30% of the cell population (Fig 4A). We then examined whether DNA damages are induced in these cells by measuring the cells with Rad52 foci, an indicator of DSB (Du et al, 2003; Matsumoto et al, 2005). Cells with nuclei containing Rad52 foci accumulated in Rif1-overexpressing cells (up to 44% of the cells at 72 h after induction; Fig 4B).

By using *GFP-α2tub* cells, we counted cells with spindle microtubules. In control cells (vector plasmid) and in the cells carrying pREP41–*Rif1*–Flag$_3$ grown with thiamine, most cells showed only cytoplasmic microtubules and roughly only 1% of cells showed spindle microtubules; either short or long spindles were detected in roughly 0.5% each of the cell population (Fig 5A). However, Rif1-overexpressing cells showed short spindles in up to 6% of the cell population and the population with long spindles decreased to

one-half of the non-induced cells (Fig 5A). This result unexpectedly suggested that at least 5–6% of cells overexpressing Rif1 arrest mitosis in the metaphase–anaphase transition. The above results suggest a possibility that the spindle assembly checkpoint is induced by Rif1 overexpression. Therefore, we examined the effects of *mad2* and *bub1* (required for SAC [Spindle Assembly Checkpoint]) mutations on the appearance of cells with spindles upon Rif1 overexpression (Bernard et al, 1998, 2001; Garcia et al, 2001; Ikui et al, 2002). The population of the cells with short spindle microtubules decreased to 1% or less in *mad2Δ* and *bub1Δ* (Fig 5B), indicating that the formation of short spindles depends on SAC. We therefore examined whether SAC is induced by overexpression of Rif1. When SAC is activated, APC/Cdc20 ubiquitin ligase is inhibited. This would stabilize Securin (Cut2) and Cyclin B. We then measured the effects of Rif1 overexpression on the duration of the Cut2 signal together with the locations of Sad1 (spindle pole body). In the control cells, the spindle appeared at 4 min from division of Sad1 foci and disappeared by 12 min. The Cut2 signal disappeared at around 14 min. In contrast, in Rif1-overproducing cells, the spindle appearing at time 0 was still visible at 36–38 min. The Cut2 signal persisted even after 30 min (Fig 5C). These results show that SAC is activated by Rif1 overexpression. We then examined the effect of SAC mutations on the growth inhibition by Rif1 overexpression. Rif1

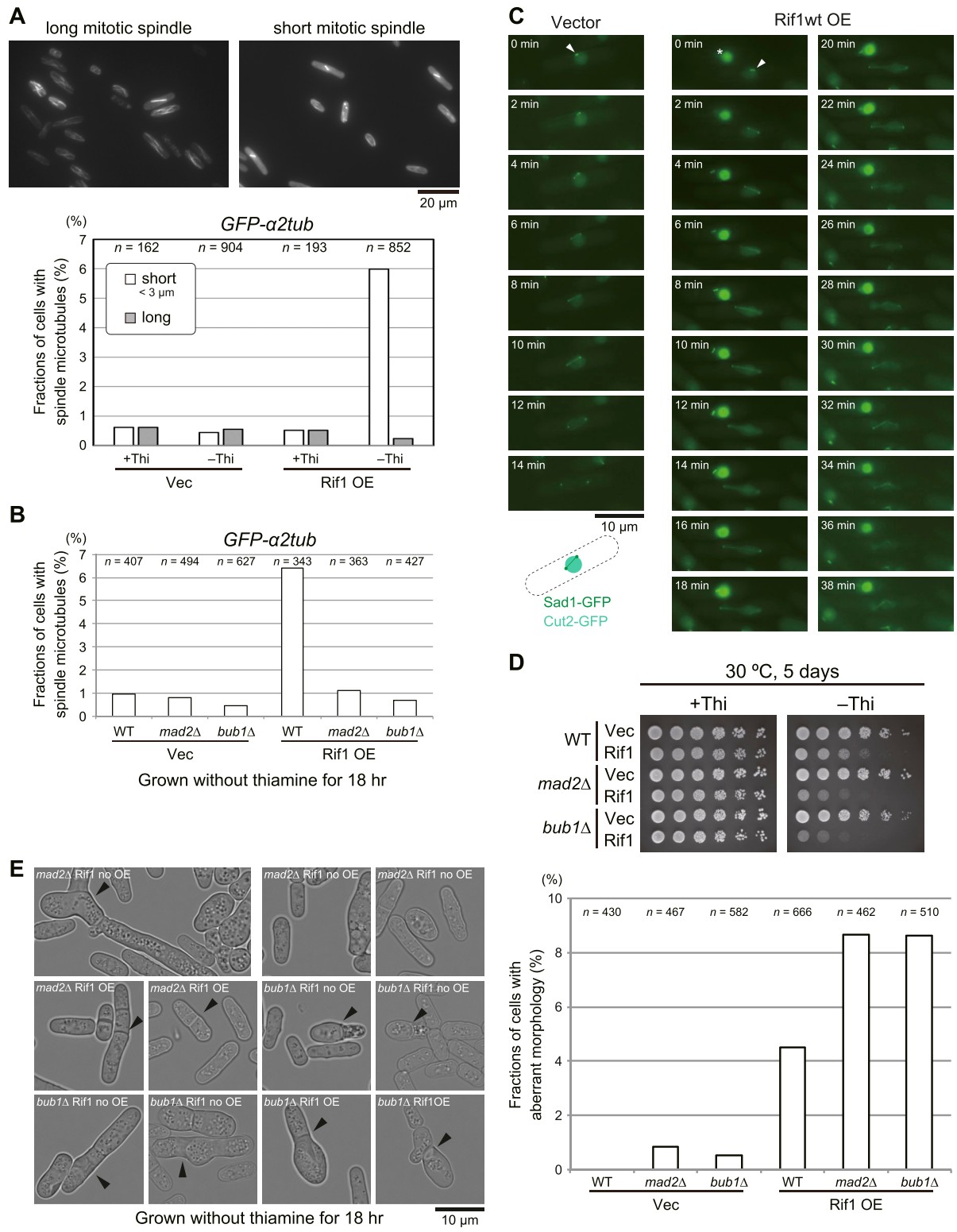

**Figure 5. Cells with short tubulin spindles are accumulated in Rif1-overexpressing cells in a manner dependent on the spindle assembly checkpoint (SAC).**
**(A)** Rif1 was overexpressed in cells expressing *GFP-α2tub* and cells with short or long spindle microtubules were counted (KYP1779 and KYP1780). In the upper panels, the photos of cells with short mitotic spindles and those with long spindles are shown. **(B)** Rif1 was overexpressed in the spindle assembly checkpoint activation mutants, *mad2Δ* or *bub1Δ*, and cells with spindle microtubules were counted (KYP1815, KYP1816, KYP1817, and KYP1818). **(C)** SAC is induced in Rif1-overproducing cells. Cells expressing Sad1–GFP (spindle pole body) and Cut2-GFP (securin) were monitored under a fluorescent microscope starting from the time when spindle pole bodies started to separate (t = 0). Mitotic spindles between the two SPB disappear and the nuclear Cut2 signal disappear in non-overproducing cells by 14 min (KYP1836), whereas

overexpression inhibited growth in *mad2Δ* and *bub1Δ* cells, indicating that growth inhibition is not caused by SAC (Fig 5D). It is of interest that Rif1 overexpression inhibited growth more vigorously in the SAC mutants than in the WT cells. Indeed, after Rif1 overexpression, cells with aberrant morphology increased to 4% in the WT cells and to 8% in *mad2Δ* and *bub1Δ* cells (Fig 5E). These results suggest that SAC activation may partially suppress the cell death–inducing effect of Rif1 overexpression.

The results indicate that the aberrant chromatin segregation is responsible for growth inhibition and cell death. We noted that cells with aberrant spindles increased in cells overexpressing Rif1, and the fractions containing these structures were greater in PP1bs mutant–overexpressing cells than in the WT Rif1-overexpressing cells (Fig S3). In contrast, the populations of the cells with short spindles decreased with the PP1bs mutant compared with the WT Rif1. This is due to the decreased level of SAC activation in the cells overexpressing PP1bs mutant than in those overexpressing the WT Rif1.

### Chromatin binding of Rif1 is necessary for growth inhibition by Rif1 overexpression, and overexpressed Rif1 induces relocation of chromatin to the nuclear periphery

We previously screened *rif1* point mutants which suppress *hsk1-89* and obtained two mutants, R236H and L848S, each of which alone could suppress *hsk1-89* (Kobayashi et al, 2019). R236H bound to Rif1bs (Rif1bs$_{I:2663}$ and Rif1bs$_{II:4255}$) and to a telomere as efficiently as the WT in chromatin immunoprecipitation (ChIP) assays. On the other hand, L848S did not bind to either of the two Rif1bs or to the telomere (Kobayashi et al, 2019). We examined the effect of overexpression of these point mutants in WT and *hsk1-89* cells. R236H which can bind to chromatin caused growth defects in both WT and *hsk1-89* when overexpressed (Fig 6A). On the other hand, L848S which is compromised in chromatin-binding activity showed very little or no growth inhibition in the WT. Interestingly, L848S inhibited the growth of *hsk1-89* (Fig 6A). Both mutant proteins were expressed at a level similar to that of the WT (data not shown).

The above results strongly suggest that chromatin binding of Rif1 is important for growth inhibition. Therefore, we have examined chromatin binding of overexpressed Rif1 protein by ChIP-seq analyses. The results indicate that overexpressed Rif1 binds to multiple sites on the chromatin, in addition to its targets in the non-overproducing WT cells (Fig 6B). 128 peaks (Rif1 no OE) and 169 peaks (Rif1 OE) were identified by peak-calling program MACS2 (listed in Tables S1 and S2) and conserved sequence motifs were identified by MEME suites from the sequences of the Rif1-binding segments. Distribution of motif position probability was determined by STREME (provided from MEME suites) (Fig 6C). G-rich motifs were conserved and distributed around the Rif1-binding

segment in both "Rif1 no OE" and "Rif1OE." Whereas there were two strong peaks on both sides of the Rif1bs summit with ~30-bp intervals in "Rif1 no OE," four peaks were detected in the 100-bp segment centering on the Rif1bs summit in "Rif1 OE," suggesting that Rif1-binding sequence specificity may be relaxed in Rif1 OE cells.

ChIP-qRT-PCR showed that overexpressed Rif1 binds to known Rif1bs sequences and to a telomere with three to seven-fold higher efficiency than the endogenous Rif1 does and binds also to a non-Rif1bs sequence (Fig 6D). These results suggest a possibility that the aberrant chromatin binding of Rif1 may be related to the induction of aberrant chromatin morphology and resulting growth inhibition and cell death.

We examined the chromatin morphology by using the cells containing GFP-labeled histone (h3.2-GFP). Interestingly, induction of Rif1 expression led to increased cell populations carrying nuclei with chromatin enriched at the nuclear periphery. This population reached over 6% with the WT and 11% with the rif1PP1BS mutant at 18 h after induction (Fig 7A and B). Enrichment of chromatin at the nuclear periphery could be caused by enlarged nucleoli as a result of Rif1 overproduction. We therefore measured the sizes of nucleoli by labeling the Gar2 protein. We did not detect any effect on the sizes of nucleoli by overexpression of the WT or PP1bs-mutant Rif1 protein (Fig S4), showing that chromatin relocation is not caused by enlarged nucleoli. The higher level of aberrant chromatin caused by the PP1bs mutant may be consistent with more severe growth inhibition with this mutant. The chromatin binding–deficient L848S mutant did not significantly induce relocation of chromatin, whereas the R236H mutant, which is capable of chromatin binding, induced the relocation in 5% of the population (Fig 7C and D), in keeping with growth-inhibiting properties of the latter mutant but not of the former.

### Nuclear dynamics of the Rif1 protein

To visualize nuclear dynamics of the Rif1 protein, we have fused a fluorescent protein to Rif1. Rif1 protein in higher eukaryotes contains a long IDP (intrinsically disordered polypeptide) segment between the N-terminal HEAT repeat sequences and C-terminal segment containing G4 binding and oligomerization activities. The fission yeast Rif1 does not carry IDP, but contains HEAT repeats and the C-terminal segment with similar biochemical activities. Thus, we speculated that insertion of a foreign polypeptide at the boundary of the two domains would least affect the overall structure of the protein and introduced the mKO2 DNA fragment at aa 1,090/1,091 or at aa 1,128/1,129 (Fig S5A). The resulting plasmid DNAs were integrated at the endogenous *rif1* locus generating MIC11-130 *rif1*–mKO2-1 and MIC12-123 *rif1*–mKO2-2, respectively.

We then evaluated the functions of the fusion proteins. *hsk1-89* (ts) cells did not grow at 30°C (non-permissive temperature), whereas

---

those in Rif1-overexpressing cells stay as late as for 38 min (KYP1837). White arrowheads indicate Sad1. The drawing shows the nuclear signals of Cut2 (pale green) and two dots of Sad2 and connecting microtubules. The strong green signals indicating by * in Rif1wt OE samples represent a dead cell. **(D)** Spot tests of SAC mutant cells overexpressing Rif1 (KYP025, KYP1805, KYP1815, KYP1816, KYP1817, and KYP1818). **(E)** Fractions of cells with aberrant morphology (indicated by arrowheads) are scored in the WT, *mad2Δ* or *bub1Δ* cells overproducing the WT Rif1 (KYP025, KYP1805, KYP1815, KYP1816, KYP1817, and KYP1818). Cells with aberrant morphology include multi-septated cells, cells with misplaced septum, enlarged cells, septated dead cells, and cells with breached morphology. Left, phase contrast images of the cells; right, quantification of cells with aberrant morphology. Rif1 OE, Rif1 overexpression. In (A, B, E), cells were grown in a medium lacking thiamine for 18 h. Source data are available for this figure.

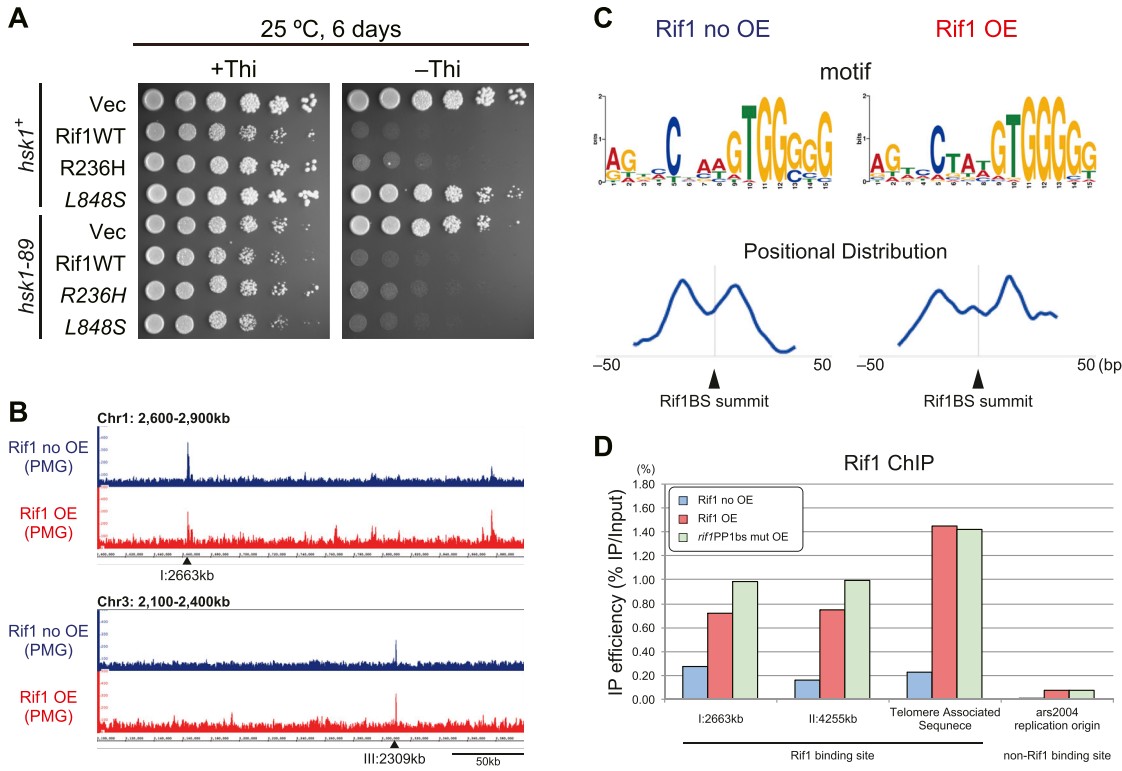

**Figure 6. Requirement of chromatin binding for growth inhibition and chromatin-binding profile of overexpressed Rif1.**
**(A)** Rif1 mutants were overexpressed in the WT (KYP1781 and KYP1782) and *hsk1-89* cells (KYP1783 and KYP1784), and spot tests were conducted. The R236H mutant binds to chromatin but the L848S mutant does not (Kobayashi et al, 2019). **(B)** KYP1268 (*nda3–KM311*, Rif1–His$_6$–Flag$_{10}$; blue) and MS733 (*nda3–KM311*, nmt1–Rif1–His$_6$–Flag$_{10}$; red) were cultured in PMG medium containing 15 μM thiamine. The cells were washed with fresh PMG medium without thiamine and grown at 30°C for 12 h. The cells were arrested at the M-phase by shifting to 19.5°C for 6 h, and then were released from the M-phase by addition of an equal volume of fresh PMG medium pre-warmed at 43°C. At 20 min after release, the cells were analyzed by ChIP-seq. Two known Rif1bs are indicated by arrowheads. **(B, C)** Motif Logo of the conserved sequence motif identified by MEME suites from the sequences of the Rif1-binding segments determined by ChIP-seq in (B), and distribution of motif position probability determined by STREME (provided from MEME suites) on the 300-bp sequences centered on the Rif1-binding summits at the 128 and 169 peaks of "Rif1 no OE" and "Rif1 OE," respectively.
**(D)** Binding of Rif1 to Rif1bs$_{I:2663kb}$, Rif1bs$_{II:4255kb}$, telomere-associated sequences (telomere of chromosome II) and ars2004 (non-Rif1bs) were measured in the WT cells harboring vector, pREP41–*Rif1–Flag$_3$*, or pREP41–*rif1PP1bsmut–Flag$_3$* by ChIP-qRT-PCR. Cells were grown in the medium lacking thiamine for 18 h before harvest. The IP efficiency was normalized by the level of input DNA.
Source data are available for this figure.

*hsk1-89 rif1Δ* cells did. On the other hand, *hsk1–89 rif1*–mKO2 did not grow at 30°C (Fig S5B), indicating that the Rif1–mKO2 retains the WT replication-inhibitory functions. To evaluate their telomere functions, we examined the telomere length in Rif1–mKO2 cells. As reported, the telomere length increased in *rif1Δ* cells (Fig S5C, lane 2), whereas that in Rif1–mKO2-1 and -2 cells did not significantly change (Fig S5C, lanes 3 and 4), suggesting that the insertion of mKO2 does not affect the Rif1 function at telomeres. We chose Rif1–mKO2 (aa 1,090/1,091) cells for further analysis. Rif1–mKO2 exhibited strong dots in nuclei (Fig S5D and E), which co-localized with Taz1-GFP or Rap1-EGFP (Fig S5D, data not shown), indicating that they represent telomeres. Minute foci appeared in nuclei, probably representing Rif1 bound to the chromosome arms. Thus, Rif1–mKO2 (aa 1,090/1,091) cells permit the visualization of dynamics of the endogenous Rif1 protein. Time lapse analyses of Rif1–mKO2 revealed a few big foci in each cell which co-localize with Taz1 along with minute other nuclear foci that are highly dynamic and represent Rif1 on chromatin arms (Video 1, Video 2, Video 3, Video 4, and Video 5 and Fig S5D).

## Overexpression of Rif1 causes relocation of the endogenous Rif1 protein

Upon overexpression of Rif1, either WT or a PP1bs mutant, in Rif1–mKO2 cells, mKO2 signals spread through nuclei (Fig S5E), consistent with the promiscuous chromatin binding of overexpressed Rif1. The overexpressed Rif1 would form mixed oligomers with the endogenous Rif1–mKO2, relocating some of the telomere-bound Rif1–mKO2 to chromosome arms. Prewash with 0.1% Triton X-100 and DNase I before PFA fixation led to appearance of multiple clear dots in nuclei in Rif1–mKO2 cells (Figs 8A and S6A) because at least some of the Rif1 bound to chromatin arms is resistant to Triton/DNaseI pretreatment and that on telomere is more sensitive. Overexpression of Rif1 resulted in increased numbers of dots (Figs 8A and B and S6A–C), consistent with the relocation of Rif1–mKO2 from the telomere to nuclear matrix-related insoluble compartments. The overall nuclear fluorescent intensities of the prewashed mKO2-Rif1 cells also increased after Rif1 overexpression compared with the vector control (Figs 8C and S6C), consistent with the above speculation. These results support the conclusion that overexpressed

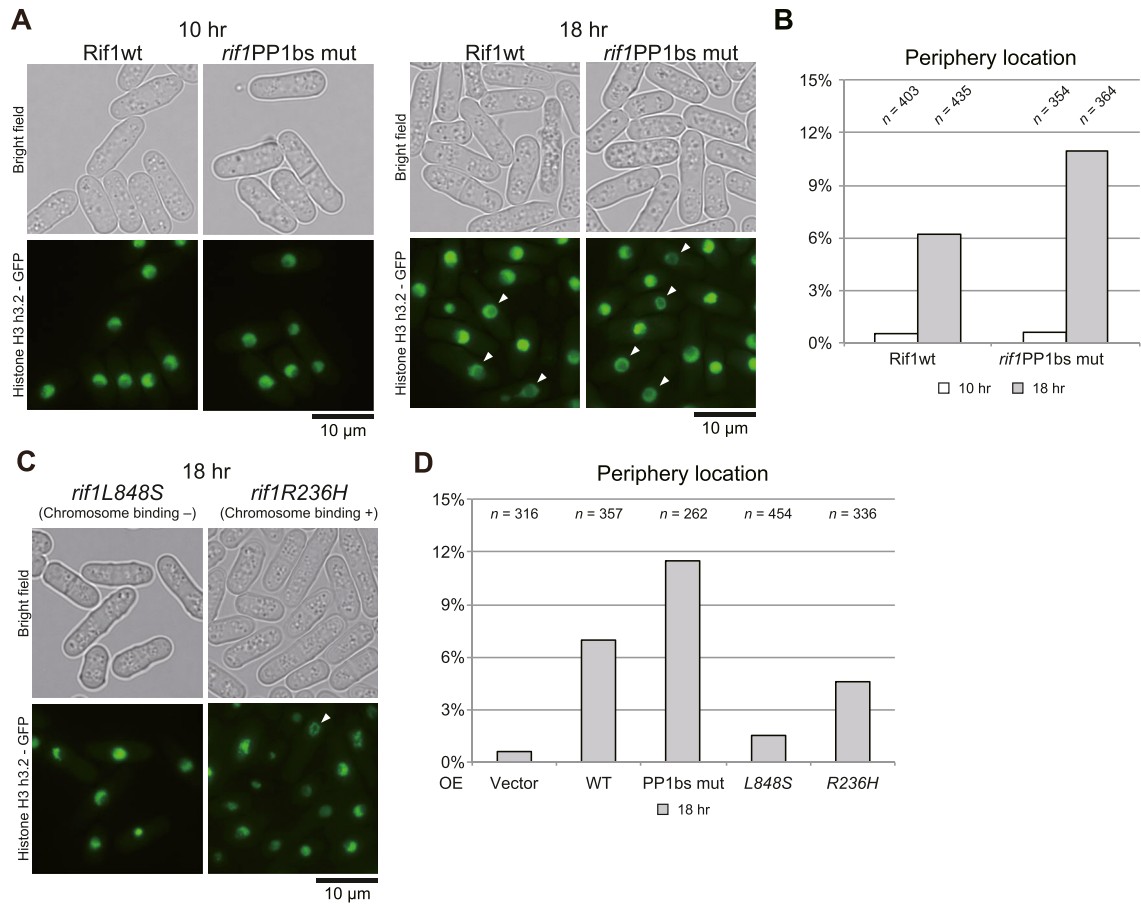

**Figure 7. Chromatin morphology of the cells after induction of Rif1 expression.**
Cells expressing GFP-fused Histone H3 (h3.2-GFP) were observed under a fluorescent microscope after induction of Rif1 protein for 10 h or 18 h, as indicated. **(A, B)** Overexpression of WT (KYP1776) or PP1bs mutant (KYP1842). **(C, D)** Overexpression of L848S (chromatin binding-deficient) (KYP1844) or R236H (chromatin binding-proficient) (KYP1843) mutant. **(A, C)** Phase contrast and fluorescent images of the cells are presented. **(A, B, C, D)** Fractions of the cells with chromatin relocated at the nuclear periphery (indicated by arrowheads in (A, C)) are calculated and presented.
Source data are available for this figure.

Rif1 promotes aberrant tethering of chromatin to nuclear membrane/nuclear matrix-related insoluble compartments and that the resulting aberrant chromatin organization causes mitotic defects.

## Discussion

Rif1 is an evolutionary conserved nuclear factor that plays roles in various chromosome transactions including DSB repair, DNA replication, transcription, and epigenetic regulation. Rif1 proteins from fission yeast and mammalian cells bind to the G4 structure and generate higher order chromatin architecture (Kanoh et al, 2015; Moriyama et al, 2018). In addition to the conserved interaction with PP1, Rif1 is known to interact with a number of proteins. *S. pombe* Rif1 interacts with telomere factors Taz1 and Rap1 (Kanoh & Ishikawa, 2001; Miller et al, 2005). This interaction is important for its function at the telomere. It also interacts with Epe1 (Wang et al, 2013), Jmjc domain chromatin-associated protein, suggesting its potential role in chromatin regulation. Human Rif1 interacts with

DSB repair factors, 53BP1, Mdc1, Bloom RecQ helicase (Feng et al, 2013; Batenburg et al, 2017; Gupta et al, 2018), and anti-silencing function 1B histone chaperone, ASF1B (Huttlin et al, 2017). This underscores its roles in the regulation of DSB repair and epigenomic state. Both fission yeast and human Rif1 are biochemically enriched in nuclear insoluble fractions, and a portion of mammalian Rif1 is localized at the nuclear periphery. It was speculated that Rif1 tethers the chromatin fiber along the nuclear membrane, generating a chromatin compartment in the vicinity of the nuclear periphery (Yamazaki et al, 2012; Kobayashi et al, 2019). However, effects of the increased level of Rif1 on chromatin localization in nuclei and its subsequent outcome have not been explored.

In this communication, we showed that increased numbers of Rif1 molecules in fission yeast cells stimulated its chromatin arm-binding and relocation of chromatin to the nuclear periphery/detergent-insoluble compartments, leading to cell death induced by aberrant mitosis. Rif1 overexpression inhibited also the S-phase, and this inhibition depends on the interaction with PP1, although PP1 was not required for chromatin relocation and growth inhibition by Rif1.

◢◢◢◢ **Life Science Alliance**

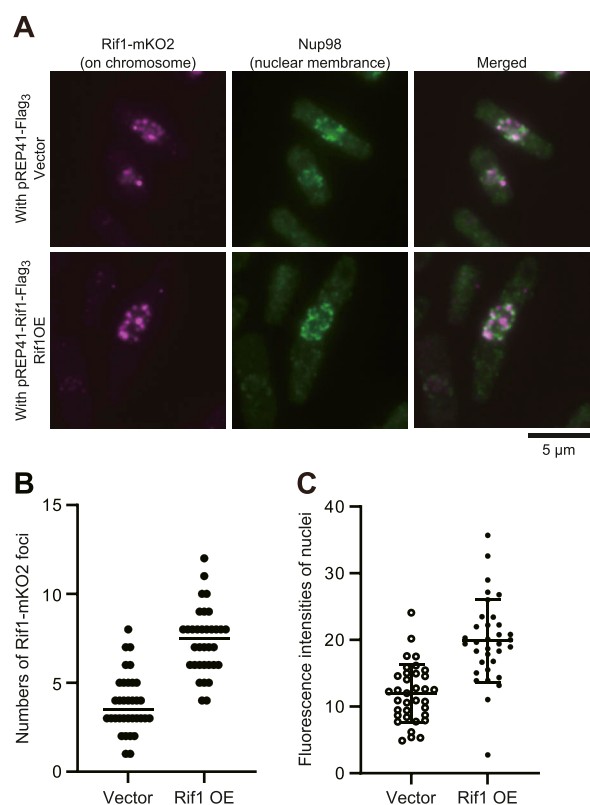

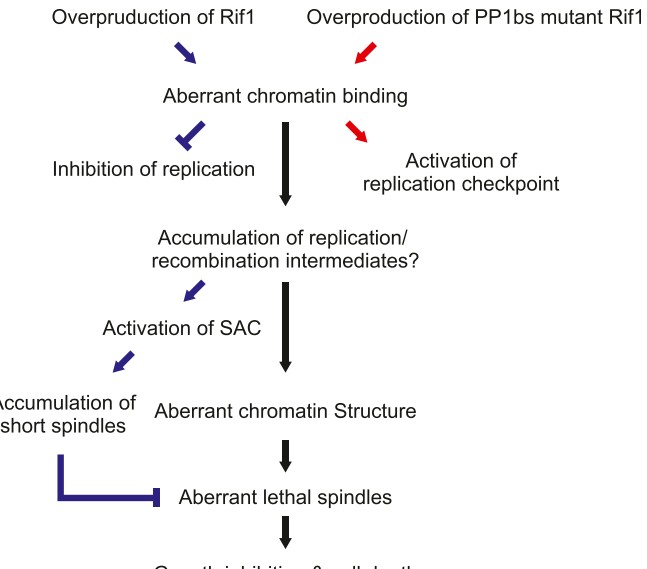

**Figure 8.  The endogenous Rif1 protein was relocated upon overexpression of Rif1.**
**(A)** Rif1–mKO2 cells (KYP1866 and KYP1867), in which the endogenous Rif1 was tagged with mKO2, harboring pREP41–$Flag_3$ vector (upper) or pREP41–$Rif1$–$Flag_3$ (lower) were grown in the absence of thiamine for 20 h, and were extracted by Triton X-100 and DNase I and remaining endogenous Rif1–mKO2 signals (mazenta) were observed. The nuclear envelope was stained with Nup98 antibody (green). **(B, C)** The numbers (B) and the intensities (C) of nuclear foci were quantified in Rif1–mKO2 cells harboring pREP41–$Flag_3$ (Vector) or pREP41–$Rif1$–$Flag_3$ (Rif1OE) grown as in (A).
Source data are available for this figure.

## Events induced by overexpression of Rif1 in fission yeast cells

We observed severe growth inhibition of fission yeast cells by overexpression of the Rif1 protein. The growth inhibition was observed only on the plate lacking thiamine where the nmt1 promoter driving the transcription of plasmid-borne Rif1 is activated. We detected ~1,000 molecules of the Rif1 protein in a growing fission yeast cell. It increased by 10 fold in cells harboring the plasmid-expressing Rif1, and upon depletion of thiamine, it increased further by 3.7 fold. Thus, the presence of Rif1 over the threshold (between ~10,000 and ~37,000 molecules per cell) may confer growth inhibition. Rif1 overexpression delayed S-phase initiation and progression in synchronized cell populations. Interestingly, S-phase inhibition was not observed by overexpression of the PP1bs mutant of Rif1, indicating that inhibition of DNA synthesis depends on the recruitment of PP1 (Fig 3A). Replication checkpoint, as measured by Cds1 kinase activity, was activated by overexpression of the PP1bs mutant of Rif1 protein, but not by the WT Rif1 protein (Fig 3C). This could be due to the ongoing S-phase in PP1bs mutant-overexpressing cells, while S-phase is

**Figure 9.  Cellular events induced by overexpression of Rif1 in fission yeast.**
Overproduction of Rif1 leads to its aberrant chromatin binding and inhibits S-phase initiation and progression through its ability to recruit PPase. Excessive chromatin binding of Rif1 results in aberrant tethering of chromatin fibers to the nuclear periphery, which may directly or indirectly inhibit proper progression of chromosome segregation, eventually leading to cell death. Overexpression of the WT Rif1 inhibits DNA replication, whereas that of PP1bs-mutant Rif1 does not inhibit DNA replication but activates the replication checkpoint. Rif1 overexpression induces SAC, leading to increased cell population with short spindles, which probably antagonizes induction of aberrant chromosome structures.

inhibited by the WT Rif1. Ongoing replication forks would be interfered by the bound Rif1 proteins, activating the replication checkpoint. On the other hand, in the WT Rif1 overexpression, initiation of DNA synthesis is blocked by Rif1, thus generating less fork blocks, that is, less replication checkpoint activation.

Cells with short spindles accumulate in Rif1-overproduced cells, and this incident depends on SAC (Fig 5B). Growth inhibition is observed in SAC mutant cells, indicating that short spindle formation is not critical for growth inhibition. In fact, severer growth inhibition was observed in the SAC mutants. Short spindles were reported to accumulate in SAC-activated cells, and it was reported that replication intermediates or recombination intermediates can activate SAC (Nakano et al, 2014), resulting in cells with short spindles. SAC was less efficiently activated by overexpression of a PP1bs mutant, consistent with reduced short spindles in this mutant. As stated above, Rif1 inhibited growth more strongly in SAC mutants than in the WT cells. The PP1bs mutant inhibited growth more strongly than the WT Rif1. These results indicate that SAC-induced mitotic arrest with short spindles may serve as a protective barrier against aberrant mitosis that would lead to cell death (Fig 9).

## Aberrant chromatin structures induced by Rif1 suggests the ability of Rif1 to tether chromatin at the nuclear periphery

Histone H3 (H3.2)-labeled chromatin is detected uniformly in the nuclei of the WT cells. Upon overexpression of Rif1 protein, nuclei

**Table 1. Strains list used in this study.**

| Strain | Genotpye | Source | Related figure |
|---|---|---|---|
| YM71 | h− leu1-32 ura4-D18 | Our Stock | Fig S5 |
| KYP025 | h− leu1-32 ura4-D18 pREP41-Flag3 | Our Stock | Figs 1C, 2A and E, 4A, 5D and E, 6A, and S2B and C |
| KYP015 | h− leu1-32 ura4-D18 pREP41-Rif1-Flag3 | Our Stock | Figs 1C, 2A and E, 4A, 5D and E, 6A, S1B, and S2B and C |
| KYP008 | h− leu1-32 ura4-D18 rif1Δ::ura4+ | Our Stock | |
| MH511 | h− leu1-32 ura4-D18 Rif1-Flag3:kanR | Our Stock | |
| KYP1827 | h− leu1-32 ura4-D18 Rif1-Flag3:kanR pREP41 | Our Stock | Figs 1A and S1A and B |
| KYP1804 | h− leu1-32 ura4-D18 rif1Δ::ura4+ pREP41 | Our Stock | Figs 1C and 2E |
| KYP1805 | h− leu1-32 ura4-D18 rif1Δ::ura4+ pREP41-Rif1-Flag3 | Our Stock | Figs 1C, 2E, and S1A |
| KYP1806 | h− leu1-32 ura4-D18 rif1Δ::ura4+ pREP41-rif1(61–1,400)-Flag3 | Our Stock | Fig 1C |
| KYP1807 | h− leu1-32 ura4-D18 rif1Δ::ura4+ pREP41-rif1(81–1,400)-Flag3 | Our Stock | Fig 1C |
| KYP1808 | h− leu1-32 ura4-D18 rif1Δ::ura4+ pREP41-rif1(151–1,400)-Flag3 | Our Stock | Figs 1C and S1A |
| KYP1853 | h− leu1-32 ura4-D18 rif1Δ::ura4+ pREP41-rif1(1–1,260)-Flag3 | Our Stock | Figs 1C and S1A |
| KYP1854 | h− leu1-32 ura4-D18 rif1Δ::ura4+ pREP41-rif1(1–965)-Flag3 | Our Stock | Figs 1C and S1A |
| KYP1855 | h− leu1-32 ura4-D18 rif1Δ::ura4+ pREP41-rif1(1–442)-Flag3 | Our Stock | Figs 1C and S1A |
| KYP1856 | h− leu1-32 ura4-D18 rif1Δ::ura4+ pREP41-rif1(443–1,400)-Flag3 | Our Stock | Figs 1C and S1A |
| KYP1857 | h− leu1-32 ura4-D18 rif1Δ::ura4+ pREP41-rif1(966–1,400)-Flag3 | Our Stock | Figs 1C and S1A |
| MS104 | h− leu1-32 ura4-D18 hsk1-89:ura4+ | Our Stock | |
| KYP1752 | h− leu1-32 ura4-D18 hsk1-89:ura4+ pREP41-Flag3 | Our Stock | Figs 1D and 6A |
| KYP1753 | h− leu1-32 ura4-D18 hsk1-89:ura4+ pREP41-Rif1-Flag3 | Our Stock | Figs 1D and 6A |
| KYP1754 | h− leu1-32 ura4-D18 hsk1-89:ura4+ pREP41-rif1(1–1,260)-Flag3 | Our Stock | Fig 1D |
| KYP1755 | h− leu1-32 ura4-D18 hsk1-89:ura4+ pREP41-rif1(1–965)-Flag3 | Our Stock | Fig 1D |
| KYP1756 | h− leu1-32 ura4-D18 hsk1-89:ura4+ pREP41-rif1(1–442)-Flag3 | Our Stock | Fig 1D |
| KYP1757 | h− leu1-32 ura4-D18 hsk1-89:ura4+ pREP41-rif1(443–1,400)-Flag3 | Our Stock | Fig 1D |
| KYP1758 | h− leu1-32 ura4-D18 hsk1-89:ura4+ pREP41-rif1(966–1,400)-Flag3 | Our Stock | Fig 1D |
| JX502 | h− dis2-11 leu1 | Our Stock | |
| JX503 | h− dis2::ura4+ leu1 ura4 | Our Stock | |
| FY9620 | h− leu1 ura4 sds21::ura4+ | NBRP | |
| KYP1759 | h− dis2-11 leu1 pREP41 | Our Stock | Fig 2A |
| KYP1760 | h− dis2-11 leu1 pREP41-Rif1-Flag3 | Our Stock | Fig 2A |
| KYP1761 | h− dis2::ura4+ leu1 ura4 pREP41 | Our Stock | Fig 2A |
| KYP1762 | h− dis2::ura4+ leu1 ura4 pREP41-Rif1-Flag3 | Our Stock | Fig 2A |
| KYP1763 | h− leu1 ura4 sds21::ura4+ pREP41 | Our Stock | Fig 2A |
| KYP1764 | h− leu1 ura4 sds21::ura4+ pREP41-Rif1-Flag3 | Our Stock | Fig 2A |
| KYP1765 | h− leu1-32 ura4-D18 pREP41 pREP42 | Our Stock | Fig 2C |
| KYP1766 | h− leu1-32 ura4-D18 pREP41 pREP42-dis2-13Myc | Our Stock | Fig 2C |
| KYP1767 | h− leu1-32 ura4-D18 pREP41 pREP42-sds21-13myc | Our Stock | Fig 2C |
| KYP1768 | h− leu1-32 ura4-D18 pREP41-Rif1-Flag3 pREP42 | Our Stock | Fig 2C |
| KYP1769 | h− leu1-32 ura4-D18 pREP41-Rif1-Flag3 pREP42-dis2-13myc | Our Stock | Fig 2C |
| KYP1770 | h− leu1-32 ura4-D18 pREP41-Rif1-Flag3 pREP42-sds21-13Myc | Our Stock | Fig 2C |
| KYP1771 | h− leu1-32 ura4-D18 pREP41-rif1PP1bs mut-Flag3 pREP42 | Our Stock | Fig 2C |
| KYP1772 | h− leu1-32 ura4-D18 pREP41-rif1PP1bs mut-Flag3 pREP42-dis2-13Myc | Our Stock | Fig 2C |

**Table 1. Continued**

| Strain | Genotpye | Source | Related figure |
|---|---|---|---|
| KYP1773 | *h− leu1-32 ura4-D18 pREP41-rif1PP1bs mut-Flag3 pREP42-sds21-13Myc* | Our Stock | Fig 2C |
| FY14160 | *h− leu1-32 ura4-D18 rif1Δ::ura4+* | NBRP | |
| KYP1839 | *h− leu1-32 ura4-D18 rif1Δ::ura4+ pREP41-rif1PP1mut-Flag3* | Our Stock | Fig 2E |
| KYP1774 | *h− leu1-32 ura4-D18 pREP41-rif1PP1mut-Flag3* | Our Stock | Figs 2E and 4A |
| KYP1268 | *h− leu1-32 ura4-D18 Rif1-His6-Flag10 nda3-KM311* | Our Stock | Figs 3A–C and 6B |
| MS733 | *h− leu1-32 ura4-D18 rif1::Pnmt-rif1-His6-Flag10:kanR nda3-KM311* | Our Stock | Figs 3A–C and 6B |
| KYP1283 | *h− leu1-32 ura4-D18 Pnmt1-rif1PP1bs mut-His6-Flag10:kanR nda3-KM311* | Our Stock | Fig 3A–C |
| FY15623 | *h− ade6-M216 leu1-32 ura4-D18 hht2+-GFP::ura4+* | NBRP | |
| KYP1775 | *h− ade6-M216 leu1-32 ura4-D18 hht2+-GFP::ura4+ pREP41* | Our Stock | Fig 4B |
| KYP1776 | *h− ade6-M216 leu1-32 ura4-D18 hht2+-GFP::ura4+ pREP41-Rif1-Flag3* | Our Stock | Fig 4B |
| MS360 | *h− leu1-32 ura4-D18 rad22-YFP:KanR* | Our Stock | |
| KYP011 | *h− leu1-32 ura4-D18 Rad52-EGFP:kanR* | Our Stock | |
| KYP1777 | *h− leu1-32 ura4-D18 rad22-YFP:KanR pREP41* | Our Stock | Fig 4C |
| KYP1778 | *h− leu1-32 ura4-D18 rad22-YFP:KanR pREP41-Rif1-Flag3* | Our Stock | Fig 4C |
| KYP1860 | *h− leu1-32 ura4-D18 Rad52-EGFP:kanR pREP41* | Our Stock | Fig 4C |
| KYP1861 | *h− leu1-32 ura4-D18 Rad52-EGFP:kanR pREP41-Rif1-Flag3* | Our Stock | Fig 4C |
| MS130 | *h+ leu1-32 ura4-D18 lys1+::pmt1-GFP-alpha2tub* | Our Stock | |
| KYP1779 | *h+ leu1-32 ura4-D18 lys1+::pmt1-GFP-alpha2tub pREP41* | Our Stock | Fig 5A and B |
| KYP1780 | *h+ leu1-32 ura4-D18 lys1+::pmt1-GFP-alpha2tub pREP41-Rif1-Flag3* | Our Stock | Figs 5A and B and S3B |
| KYP1801 | *h− leu1-32 ura4-D18 mad2::ura4+ lys1+::pmt1-GFP-alpha2tub* | Our Stock | |
| KYP1815 | *h− leu1-32 ura4-D18 mad2::ura4+ lys1+::pmt1-GFP-alpha2tub rif1Δ::hphMX6 pREP41-Flag3* | Our Stock | Fig 5B, D, and E |
| KYP1816 | *h− leu1-32 ura4-D18 mad2::ura4+ lys1+::pmt1-GFP-alpha2tub rif1Δ::hphMX6 pREP41-Rif1-Flag3* | Our Stock | Fig 5B, D, and E |
| KYP1802 | *h− ade6-M216 leu1-32 ura4-D18 bub1::ura4+ lys1+::pmt1-GFP-alpha2tub* | Our Stock | |
| KYP1817 | *h− ade6-M216 leu1-32 ura4-D18 bub1::ura4+ lys1+::pmt1-GFP-alpha2tub rif1Δ::hphMX6 pREP41-Flag3* | Our Stock | Fig 5B, D, and E |
| KYP1818 | *h− ade6-M216 leu1-32 ura4-D18 bub1::ura4+ lys1+::pmt1-GFP-alpha2tub rif1Δ::hphMX6 pREP41-Rif1-Flag3* | Our Stock | Fig 5B, D, and E |
| 23-B10 | *h90 ade6-M216 leu1 his3-D1 cut2-GFP << kanR sad1-GFP << kanR* | Gifted from Dr. Ueno | |
| KYP1836 | *h90 ade6-M216 leu1 his3-D1 cut2-GFP << kanR sad1-GFP << kanR pREP41-Flag3* | Our Stock | Fig 5C |
| KYP1837 | *h90 ade6-M216 leu1 his3-D1 cut2-GFP << kanR sad1-GFP << kanR pREP41-Rif1-Flag3* | Our Stock | Fig 5C |
| KYP1781 | *h− leu1-32 ura4-D18 pREP41-rfi1R236H-Flag3* | Our Stock | Fig 6A |
| KYP1782 | *h− leu1-32 ura4-D18 pREP41-rfi1L848S-Flag3* | Our Stock | Fig 6A |
| KYP1783 | *h− leu1-32 ura4-D18 hsk1-89:ura4+ pREP41-rfi1R236H-Flag3* | Our Stock | Fig 6A |
| KYP1784 | *h− leu1-32 ura4-D18 hsk1-89:ura4+ pREP41-rfi1L848S-Flag3* | Our Stock | Fig 6A |
| KYP1842 | *h− ade6-M216 leu1-32 ura4-D18 hht2+-GFP::ura4+ pREP41-rif1PP1mut-Flag3* | Our Stock | Fig 7A |
| KYP1843 | *h− ade6-M216 leu1-32 ura4-D18 hht2+-GFP::ura4+ pREP41-rif1R236H-Flag3* | Our Stock | Fig 7C |
| KYP1844 | *h− ade6-M216 leu1-32 ura4-D18 hht2+-GFP::ura4+ pREP41-rif1L848S-Flag3* | Our Stock | Fig 7C |
| MS742 | *h− leu1-32 ura4-D18 rif1:mKO2 cut11-GFP-ura4+* | Our Stock | |
| KYP1866 | *h− leu1-32 ura4-D18 rif1:mKO2 cut11-GFP-ura4+ pREP41-Flag3* | Our Stock | Figs 8A and S5E |
| KYP1867 | *h− leu1-32 ura4-D18 rif1:mKO2 cut11-GFP-ura4+ pREP41-Rif1-Flag3* | Our Stock | Figs 8A and S5E |

| Strain | Genotpye | Source | Related figure |
|---|---|---|---|
| MS580 | h− leu1-32 ura4-D18 rif1::Pnmt1-rif1:kanR | Our Stock | |
| KYP1785 | h− leu1-32 ura4-D18 pREP42 | Our Stock | Fig S2A |
| KYP1786 | h− leu1-32 ura4-D18 pREP42-hsk1-him1 | Our Stock | Fig S2A |
| KYP1787 | h− leu1-32 ura4-D18 rif1Δ::ura4+ pREP42 | Our Stock | Fig S2A |
| KYP1788 | h− leu1-32 ura4-D18 rif1Δ::ura4+ pREP42-hsk1-him1 | Our Stock | Fig S2A |
| KYP1789 | h− leu1-32 ura4-D18 rif1::Pnmt1-rif1:kanR pREP42 | Our Stock | Fig S2A |
| KYP1790 | h− leu1-32 ura4-D18 rif1::Pnmt1-rif1:kanR pREP42-hsk1-him1 | Our Stock | Fig S2A |
| FY14161 | h− leu1-32 ura4-D18 taz1::ura4+ | NBRP | |
| MS129 | h− leu1-32 ura4-D18 tel1-D1::kanMX4 rad3::ura4+ | Our Stock | |
| NI392 | h− ade6-M216 leu1-32 ura4-D18 rad3::ura4+ | Our Stock | |
| MS221 | h+ ade6-M216 leu1-32 ura4-D18 chk1::ura4+ | Our Stock | |
| MS290 | h− leu1-32 ura4-D18 tel1-D1::kanMX4 | Our Stock | |
| NI453 | h− leu1-32 ura4-D18 cds1::ura4+ | Our Stock | |
| FY7826 | h− leu1-32 ura4-D18 mad2::ura4 | NBRP | |
| MS182 | h− leu1-32 ura4-D18 cdc25-22 | Our Stock | |
| MS195 | h− ade6-M216 leu1-32 ura4-D18 wee1-50 | Our Stock | |
| KYP1875 | h− leu1-32 ura4-D18 taz1::ura4+ pREP41 | Our Stock | Fig S2B |
| KYP1876 | h− leu1-32 ura4-D18 taz1::ura4+ pREP41-Rif1-Flag3 | Our Stock | Fig S2B |
| KYP1877 | h− leu1-32 ura4-D18 tel1-D1::kanMX4 rad3::ura4+ pREP41 | Our Stock | Fig S2B |
| KYP1878 | h− leu1-32 ura4-D18 tel1-D1::kanMX4 rad3::ura4+ pREP41-Rif1-Flag3 | Our Stock | Fig S2B |
| KYP1879 | h− ade6-M216 leu1-32 ura4-D18 rad3::ura4+ pREP41 | Our Stock | Fig S2B |
| KYP1880 | h− ade6-M216 leu1-32 ura4-D18 rad3::ura4+ pREP41-Rif1-Flag3 | Our Stock | Fig S2B |
| KYP1881 | h+ ade6-M216 leu1-32 ura4-D18 chk1::ura4+ pREP41 | Our Stock | Fig S2B |
| KYP1882 | h+ ade6-M216 leu1-32 ura4-D18 chk1::ura4+ pREP41-Rif1-Flag3 | Our Stock | Fig S2B |
| KYP1883 | h− leu1-32 ura4-D18 tel1-D1::kanMX4 pREP41 | Our Stock | Fig S2B |
| KYP1884 | h− leu1-32 ura4-D18 tel1-D1::kanMX4 pREP41-Rif1-Flag3 | Our Stock | Fig S2B |
| KYP1885 | h− leu1-32 ura4-D18 cds1::ura4+ pREP41 | Our Stock | Fig S2C |
| KYP1886 | h− leu1-32 ura4-D18 cds1::ura4+ pREP41-Rif1-Flag3 | Our Stock | Fig S2C |
| KYP1887 | h− leu1-32 ura4-D18 mad2::ura4 pREP41 | Our Stock | Fig S2C |
| KYP1888 | h− leu1-32 ura4-D18 mad2::ura4 pREP41-Rif1-Flag3 | Our Stock | Fig S2C |
| KYP1889 | h− leu1-32 ura4-D18 cdc25-22 pREP41 | Our Stock | Fig S2C |
| KYP1890 | h− leu1-32 ura4-D18 cdc25-22 pREP41-Rif1-Flag3 | Our Stock | Fig S2C |
| KYP1891 | h− ade6-M216 leu1-32 ura4-D18 wee1-50 pREP41 | Our Stock | Fig S2C |
| KYP1892 | h− ade6-M216 leu1-32 ura4-D18 wee1-50 pREP41-Rif1-Flag3 | Our Stock | Fig S2C |
| KYP1847 | h+ leu1-32 ura4-D18 lys1+::pmt1-GFP-alpha2tub rif1Δ21-1400AA::hphMX6 pREP41-rif1PP1mut-Flag3 | Our Stock | Fig S3B |
| KYP1863 | h− ade6-M216 leu1-32 ura4-D18 hht2+-GFP::ura4+ Gar2-mCherry:hphMX6 pREP41-Flag3 | Our Stock | Fig S4A |
| KYP1864 | h−− ade6-M216 leu1-32 ura4-D18 hht2+-GFP::ura4+ Gar2-mCherry:hphMX6 pREP41-Rif1-Flag3 | Our Stock | Fig S4A |
| KYP1865 | h− ade6-M216 leu1-32 ura4-D18 hht2+-GFP::ura4+ Gar2-mCherry:hphMX6 pREP41-rif1PP1bs mut-Flag3 | Our Stock | Fig S4A |
| MIC2-11 | h− leu1-32 ura4-D18 rif1:mKO2 | Our Stock | Fig S5A–C |
| MS744 | h− leu1-32 ura4-D18 rif1:mKO2 hsk1-89:ura4+ | Our Stock | Fig S5A–C |

| Strain | Genotpye | Source | Related figure |
|--------|----------|--------|----------------|
| HM214 | h+ leu1-32 ura4-D16 rif1::ura4+ hsk1-89:ura4+ | Our Stock | Fig S5B |
| MIC20-42 | h− leu1-32 ura4-D18 taz1-GFP::kanMX, rif1:mKO2,nda3-KM311 | Our Stock | Fig S5D and Video 1, Video 2, Video 3, Video 4, and Video 5 |
| KYP1868 | h− leu1-32 ura4-D18 rif1:mKO2 cut11-GFP-ura4+ pREP41-rif1PP1mut-Flag3 | Our Stock | Fig S5E |

with chromatin enriched at the nuclear periphery were observed in more than 6% or 10% of the cells with the WT or PP1bs mutant Rif1, respectively. Furthermore, chromatin binding-deficient mutant, L848S, failed to induce chromatin relocation to the nuclear periphery (Fig 7).

By using a Rif1 derivative containing mKO2 between the HEAT repeat and C-terminal segment, cellular Rif1 dynamics was examined. In addition to the strong dots corresponding to telomeres (Fig S5D and E), fine dots representing arm binding were observed ([Klein et al, 2021] Video 5). Upon overexpression of Rif1, the endogenous Rif1-derived mKO2 signals relocated from telomeres to the entire areas of the nuclei (Fig S5E), indicating that overexpressed Rif1, forming multimers with Rif1–mKO2, spreads over the chromosome arms. The prewash of nuclei with detergent and DNase I enhanced the Rif1 signals in nuclei, notably at the nuclear periphery (Fig S6B). Both the numbers of foci and overall intensities of nuclear signals increased upon Rif1 overproduction (Fig 8B and C), presumably due to relocation of endogenous Rif1 from telomere to chromosome arms and to detergent- and DNase I-resistant insoluble compartments through mixed oligomer formation with the overexpressed Rif1.

Our results are consistent with the idea that Rif1 promotes association of chromatin with detergent-insoluble membrane fractions, which are known to be enriched with S-acylated proteins (Hooper, 1999). Like budding yeast Rif1, fission yeast Rif1 may also be S-acylated (Fontana et al, 2019).

### Mechanisms of formation of aberrant microtubule spindles

Rif1 overexpression ultimately induces cell death through aberrant mitosis. In addition to cells with short spindles, those with aberrant defective microtubules appear. The fractions of these cells increase in PP1 mutant cells and in PP1bs mutant–overproducing cells. PP1–Rif1 interaction is regulated by the phosphorylation of Rif1 and Aurora B kinase which was reported to play a major role in phosphorylating the PP1bs, promoting the dissociation of PP1 from Rif1 during the M-phase (Nasa et al, 2018; Bhowmick et al, 2019). Overexpressed Rif1 would recruit PP1, counteracting the phosphorylation events by Aurora B and other kinases essential for mitosis. This may lead to misconduct in mitotic events. However, because PP1bs mutants can also cause aberrant microtubules and cell death, PP1 may not be the primary cause for aberrant microtubule cell death. The aberrant chromatin structure caused by overexpressed Rif1 may affect the mitotic chromatin structure, leading to mitotic defect. Alternatively, the aberrant association of Rif1 with mitotic kinases and potentially with microtubules could directly be linked to deficient microtubules in Rif1-overproducing cells.

In summary, overexpression of Rif1 would lead to relocation of the chromatin segment located in the interior of nuclei (early-replicating loci) to the nuclear periphery. Replication of DNA in the vicinity of tethered chromatin segment would be inhibited upon recruitment of PP1. It was previously reported that artificial tethering of an early-firing origin at the nuclear periphery did not render it late-firing in budding yeast (Ebrahimi et al, 2010). This is consistent with our result that Rif1-mediated chromatin recruitment at nuclear membrane alone does not inhibit the S-phase, and that the recruitment of PP1 by Rif1 is required for the inhibition. On the other hand, recruitment of chromatin to the nuclear periphery by Rif1 is sufficient to cause an aberrant M-phase and eventually cell death. The results described in this report support the idea that Rif1-mediated chromatin association with the nuclear periphery needs to be precisely regulated for coordinated progression of S- and M-phases. However, we cannot rule out the possibility that the phenotypes we observe upon Rif1 overexpression could be secondary consequences of its effect on transcription or on other chromosomal events including repair and recombination. More detailed studies will be needed to precisely determine the effects of deregulated chromatin association with nuclear membrane on cell cycle progression and cell survival.

## Materials and Methods

### Medium for *S. pombe*

YES medium contains 0.5% yeast extract (Gibco), 3% glucose (FUJIFILM Wako), and 0.1 mg/ml each of adenine (Sigma-Aldrich), uracil (Sigma-Aldrich), leucine (FUJIFILM Wako), lysine (FUJIFILM Wako), and histidine (FUJIFILM Wako). YES plates were made by adding 2% agar (Gibco) to the YES medium. Synthetic dextrose minimal medium (SD) contains 6.3 g/liter Yeast Nitrogen Base w/o Amino Acids (BD), 2% glucose, and 0.1 mg/ml each of the required amino acids. Edinburgh Minimal Medium (EMM) contains 12.3 g/liter EMM Broth without Nitrogen (Formedium), 2% glucose, and 0.1 mg/ml each of required amino acids. Pombe Minimal Glutamate (PMG) contains 27.3 g/liter EMM Broth without Nitrogen, 5 g/liter L-glutamic acid (Sigma-Aldrich), and 0.1 mg/ml each of required amino acids. 15 $\mu$M thiamine (Sigma-Aldrich) was added to EMM or PMG medium to repress the nmt1 promoter activity. Yeast strains and plasmids used in this study are listed in Tables 1 and 2.

### Synchronization and cell cycle analysis by flow cytometry

Rif1 expression in the yeasts containing *nda3-KM311* mutation (KYP1268, MS733, and KYP1283) was inducted for 12 h in the PMG medium without thiamine. The yeasts were arrested at 20°C for 6 h and then the cells were synchronized at the M-phase and Rif1

**Table 2. Oligonucleotides & Plasmids.**

| Oligonucleotides | Sequence |
|---|---|
| rif1-fw-1 | 5′-CTTTGTTAAATCATATGACAAAAGAAATTGCTGTGAAGGAGGCT-3′ |
| rif1-fw-151 | 5′-CTTTGTTAAATCATATGTTATCGGATAGATGCTCTAACAATTCAGAG-3′ |
| rif1-fw-443 | 5′-CTTTGTTAAATCATATGACTACTTTGATTGCTTTAATATATGCA-3′ |
| rif1-fw-966 | 5′-CTTTGTTAAATCATATGTCCACTGCTACAGCTTCTAATATTTTAGAA-3′ |
| pREP_rif12_60_Fwd | 5′-AAATCATATGCCCAAACGAGGTATCTTAAAAACTTCAACAC-3′ |
| pREP_rif12_80_Fwd | 5′-AAATCATATGCAATCCTTTGAAGGAAATGAATCTG-3′ |
| rif-rv-1400 | 5′-TCTAGAGTCGACATAAGCAATTCTAGATAAAATAGCTCTCTGTAA-3′ |
| rif-rv-1260 | 5′-TCTAGAGTCGACATAAACTTCCTTATTCACGTTGGAAGATTGGCT-3′ |
| rif-rv-965 | 5′-TCTAGAGTCGACATAATTTAGTAGCAGCTGCAAAATTAATATAC-3′ |
| rif-rv-442 | 5′-TCTAGAGTCGACATATGCTGCATTCTTTACTGTTGGCAAATTCC-3′ |
| pREP_rif1_2_60_Rev | 5′-TTGGGCATATGATTTAACAAAGCGACTATAAGTCAGAAAG-3′ |
| pREP_rif1_2_80_Rev | 5′-GATTGCATATGATTTAACAAAGCGACTATAAGTCAGAAAG-3′ |
| Fusion-dis2N | 5′-CTTTGTTAAATCATATGTCGAACCCAGATGTGGATTTGGATTCC-3′ |
| dis2MycFusion | 5′-ATTAACCCGGGGATCAACTTTGAATTTCCTGTCTTATTCTTCCGAGG-3′ |
| Fusion-sds21 | 5′-CTTTGTTAAATCATATGGATTATGATATTGATGCGATTATTGAA-3′ |
| sds21MycFusion | 5′-ATTAACCCGGGGATCAAATTATTTTTGGATTTCTTCAAACTGTTCGT-3′ |
| Rif1_PP1_Mut1_Fwd | 5′-GCTGCCAAGAAGGCAAATGCTAGTAGTGAA-3′ |
| Rif1_PP1_Mut1_Rev | 5′-GCATTTGCCTTCTTGGCAGCAATCGAACTA-3′ |
| Rif1_PP1_Mut2_Fwd | 5′-CCCAAACGAGGTGCCGCAAAAACTTCAACA-3′ |
| Rif1_PP1_Mut2_Rev | 5′-GAAGTTTTTGCGGCACCTCGTTTGGGTAAA-3′ |
| **Recombinant DNA (Plasmids)** | |
| pREP41 | |
| pREP41-*Flag3* | |
| pREP41-*Rif1-Flag3* | |
| pREP41-*rif1(1–1,260)-Flag3* | |
| pREP41-*rif1(1–965)-Flag3* | |
| pREP41-*rif1(1–442)-Flag3* | |
| pREP41-*rif1(151–1,400)-Flag3* | |
| pREP41-*rif1(443–1,400)-Flag3* | |
| pREP41-*rif1(966–1,400)-Flag3* | |
| pREP41-*rif1(61–1,400)-Flag3* | |
| pREP41-*rif1(81–1,400)-Flag3* | |
| pREP42 | |
| pREP42-*dis2-myc13* | |
| pREP42-*dis2-myc13* | |
| pREP42-*sds21-myc13* | |
| pREP42-*sds21-myc13* | |
| pREP41-*rif1PP1bs mut-Flag3* | |
| pREP41-*rif1R236H-Flag3* | |
| pREP41-*rif1L848S-Flag3* | |
| pREP42-Dual-*Hsk1-Him1* | |

expression was inducted for 18 h. They were released into a subsequential cell cycle at 30°C. Cells in 5 ml culture were collected and suspended in 200 μl water. Cells were fixed with 600 μl ethanol, washed with 50 mM sodium citrate (pH 7.5) (FUJIFILM Wako), and were treated with 0.1 mg/ml RNase A (Sigma-Aldrich) in 300 μl of 50 mM sodium citrate at 37°C for 2 h. Cells were stained with 4 ng/ml propidium iodide (Sigma-Aldrich) at room temperature for 1 h. After sonication, cells were analyzed by BD LSRFortessaTM X-20.

## Co-immunoprecipitation

The procedure was performed as described previously (Shimmoto et al, 2009). For immunoprecipitation, ~1.0 × 10$^8$ cells from 50 ml culture were harvested and washed once with PBS. The cells were then re-suspended in 0.5 ml of IP buffer (20 mM HEPES-KOH [pH 7.6] [Nacalai tesque], 50 mM potassium acetate [Sigma-Aldrich], 5 mM magnesium acetate [FUJIFILM Wako], 0.1 M sorbitol [FUJIFILM Wako], 0.1% TritonX-100 [Sigma-Aldrich], 2 mM DTT [FUJIFILM Wako], 20 mM Na$_3$VO$_4$ [Sigma-Aldrich], 50 mM β-glycerophosphate [Sigma-Aldrich], and Protease Inhibitor Cocktail [Sigma-Aldrich]) and were disrupted with glass beads using a multi-beads shocker (Yasui Kikai). The lysates were cleared by centrifugation (20,000$g$ for 10 min at 4°C). The supernatants of lysates were mixed with anti-c-Myc antibody (Nacalai tesque) attached to Protein G Dynabeads (10004D; Thermo Fisher Scientific). After incubating for 1 h, the beads were washed with IP buffer and proteins were extracted by boiling with 1× sample buffer (2% SDS [Nacalai Tesque], 4 M Urea [Nacalai Tesque], 60 mM Tris–HCl [pH 6.8] [Nacalai Tesque], 10% Glycerol [Nacalai Tesque], and 70 mM 2-mercaptethanol [Sigma-Aldrich]).

## Immunoblot

Protein samples and prestained molecular weight markers (Bio-Rad) were loaded onto 5–20% gradient precast PAGE gel (ATTO corp.) and transferred to PVDF membranes (Millipore). The membranes were blocked with 5% skim milk in TBST and target proteins were detected with ANTI-FLAG M2 antibody (Sigma-Aldrich) and anti-α-Tubulin (SantaCruz).

## ChIP

1.0 × 10$^9$ cells were cross-linked with 1% formaldehyde for 15 min at 30°C and prepared for ChIP as previously described (Katou et al, 2003; Kanoh et al, 2015). Briefly, cross-linked cell lysates prepared by the multi-beads shocker (Yasui Kikai Co.) and sonication were incubated with Protein G Dynabeads (10004D; Thermo Fisher Scientific) attached to ANTI-FLAG M2 antibody (Sigma-Aldrich) for 4 h at 4°C. The beads were washed several times and the precipitated materials were eluted by incubation in elution buffer (50 mM Tris–HCl [pH 7.6], 10 mM EDTA, and 1% SDS) for 20 min at 68°C. The eluates were incubated at 68°C overnight to reverse crosslinks and then treated with RNaseA (Sigma-Aldrich) and proteinase K (FUJIFILM Wako). DNA was precipitated with ethanol in the presence of glycogen (Nacalai tesque) and further purified by using QIAquick PCR purification kit (QIAGEN).

## Living cell analysis

Cells were observed on BZ-X700 (KEYENCE) equipped with Nikon PlanApoλ 100× (NA = 1.45) using IMMERSION OIL TYPE NF2 (Nikon). Mitotic spindles were visualized by expressing Pmt1-GFP-α-Tubulin. DNA damages were detected by observing fluorescent Rad52 foci (EGFP or YFP). Securin and spindle pole bodies were visualized by expressing Cut2-GFP and Sad1-GFP, respectively. The time-lapse images were observed on PMG medium/2% agarose (Nacalai tesque). Whole chromosome locations were visualized by expressing hht2 (Histone H3 h3.2)-GFP.

## Next-generation sequencing and ChIP-Seq

Next-generation sequencing libraries were prepared as described previously (Kanoh et al, 2015). The input and the immunoprecipitated DNAs were fragmented to an average size of ~150 bp by ultra-sonication (Covaris). The fragmented DNAs were end-repaired, ligated to sequencing adapters, and amplified using NEBNext Ultra II DNA Library Prep Kit for Illumina and NEBNext Multiplex Oligos for Illumina (New England Biolabs). The amplified DNA (around 275 bp in size) was sequenced on Illumina MiSeq to generate single reads of 100 bp. The generated ChIP or input sequences were aligned to the *S. pombe* genomic reference sequence provided from PomBase by Bowtie 1.0.0 using default settings. Peaks were called with model-based analysis of ChIP-Seq (MACS2.0.10) using the following parameters: macs2 callpeak -t ChIP.sam -c Input.sam -f SAM -g 1.4e10$^7$ -n result_file −B -q 0.01. The pileup graphs were loaded on Affymetrix Integrated Genome Browser (IGB 8.0). To identify consensus conserved sequences for Rif1 binding, 300-bp DNA segments around the summits of the 128 or 169 Rif1bs identified by MACS2 were extracted and analyzed by MEME suite (Bailey et al, 2015).

## In-gel kinase assay

In-gel kinase assays for replication checkpoint activation were conducted as described previously (Geahlen et al, 1986; Waddell et al, 1995; Takeda et al, 2001). SDS–polyacrylamide gel (10%) was cast in the presence of 0.5 mg/ml myelin basic protein (Sigma-Aldrich) within the gel. Extracts (100 μg of protein) prepared by the boiling method were run on the gel. After electrophoresis, the gel was washed successively in 50 mM Tris–HCl (pH 8.0), 50 mM Tris–HCl (pH 8.0) +5 mM 2-mercaptoethanol, and denatured in 6 M guanidium hydrochloride (Nacalai tesque) in 50 mM Tris–HCl (pH 8.0) +5 mM 2-mercaptoethanol, and renatured in 50 mM Tris, pH 8.0 +5 mM 2-mercaptoethanol +0.04% Tween 20 over 12–18 h at 4°C. The gel was then equilibrated in the kinase buffer containing 40 mM HEPES-KOH (pH 7.6), 40 mM potassium glutamate, 5 mM magnesium acetate, 2 mM dithiothreitol, and 0.1 mM EGTA for 1 h at room temperature, and was incubated in the same kinase buffer containing 5 μM ATP and 50 μCi of [γ-$^{32}$P]ATP for 60 min at room temperature, followed by extensive washing in 5% trichloroacetic acid (Nacalai tesque) +1% sodium pyrophosphate until no radioactivity is detected in the washing buffer. The gel was dried and auto-radiographed.

**Table 3.  Reagents & Resources.**

| Reagent and resource | | |
|---|---|---|
| **Antibody** | | |
| Mouse anti-Flag(M2) | Sigma-Aldrich | Cat# F1804 |
| α Tubulin | SANTA CRUZ BIOTECHNOLOGY, INC. | Cat# sc-23948 |
| Peroxidase AffiniPure F(ab')₂ Fragment Donkey Anti-Mouse IgG (H+L) | Jackson Immune Research | Cat# 715-036-151 |
| Peroxidase AffiniPure F(ab')₂ Fragment Donkey Anti-Rabbit IgG (H+L) | Jackson Immune Research | Cat# 711-036-152 |
| c-Myc(A-14) | SANTA CRUZ BIOTECHNOLOGY, INC. | Cat# sc-789 |
| Anti-c-Myc(Mouse IgG1-κ), Monoclonal(MC045), AS | nacalai tesque | Cat# 04362-34 |
| Anti-Nup98 antibody, rat monoclone (2H10) | Bioacademia | Cat# 70-310 |
| **Chemicals, peptide, and recombinant proteins** | | |
| Bact Yeast Extract | Gibco | Cat# 212750 |
| Difco Yeast Nitrogen Base w/o Amino Acids | BD | Cat# 291940 |
| Bacto Agar | Gibco | Cat# 214010 |
| D(+)-Glucose | FUJIFILM Wako Pure Chemical Corporation | Cat# 049-31165 |
| Adenine hemisulfate salt | Sigma-Aldrich | Cat# A9126 |
| Uracil | Sigma-Aldrich | Cat# U0750 |
| L-Leucine | FUJIFILM Wako Pure Chemical Corporation | Cat# 124-00852 |
| D(+)-Lysine Monohydrochloride | FUJIFILM Wako Pure Chemical Corporation | Cat# 121-01461 |
| L-Histidine | FUJIFILM Wako Pure Chemical Corporation | Cat# 084-00682 |
| Thiamin Hydrochloride | FUJIFILM Wako Pure Chemical Corporation | Cat# 201-00852 |
| EMM BROTH WITHOUT NITROGEN | FORMEDIUM | Cat# PMD1302 |
| EMM BROTH WITHOUT DEXTROSE | FORMEDIUM | Cat# PMD0402 |
| L-Glutamic acid monosodium salt hydrate | Sigma-Aldrich | Cat# G5889 |
| Trisodium Citrate Dihydrate | FUJIFILM Wako Pure Chemical Corporation | Cat# 191-01785 |
| Propidium iodide | Sigma-Aldrich | Cat# P4170 |
| Ribonuclease A from bovine pancreas Type II-A | Sigma-Aldrich | Cat# R5000 |
| HEPES[N-(2-Hydroxyethyl)piperazine-N'-2-ethanesulfonic Acid] | nacalai tesque | Cat# 17514-15 |
| Potassium acetate | Sigma-Aldrich | Cat# P1190 |
| Magnesium Acetate Tetrahydrate | FUJIFILM Wako Pure Chemical Corporation | Cat# 130-00095 |
| Protease Inhibitor Cocktail for use with fungal and yeast extracts | Sigma-Aldrich | Cat# P8215 |
| D-Glucitol | nacalai tesque | Cat# 32021-95 |
| Triton X-100 | Sigma-Aldrich | Cat# T9284 |
| β-Glycerophosphate disodium salt hydrate | Sigma-Aldrich | Cat# G5422 |
| Sodium orthovanadate | Sigma-Aldrich | Cat# S6508 |
| (+/−)-Dithiothreitol | FUJIFILM Wako Pure Chemical Corporation | Cat# 042-29222 |
| Dynabeads Protein G | Thermo Fisher Scientific | Cat# DB10004 |
| Tris(hydroxymethyl)aminomethane | nacalai tesque | Cat# 35434-21 |
| EDTA 2Na Dihydrate | nacalai tesque | Cat# 15130-95 |
| Sodium Lauryl Sulfate (SDS) | nacalai tesque | Cat# 31607-65 |
| Urea | nacalai tesque | Cat# 35940-65 |
| Bromophenol Blue | FUJIFILM Wako Pure Chemical Corporation | Cat# 029-02912 |

| Reagent and resource | | |
|---|---|---|
| 2-Mercaptoethanol | Sigma-Aldrich | Cat# M7522 |
| Glycerol | nacalai tesque | Cat# 17018-83 |
| Polyoxyethylene Sorbitan Monolaurate (Tween 20) | nacalai tesque | Cat# 28353-85 |
| IGEPAL CA-630 | Sigma-Aldrich | Cat# I8896 |
| Precision Plus Protein Dual Color Standards | Bio-Rad | Cat# 1610394 |
| Immobilon-P PVDF Membrane | Millpore | Cat# IPVH00010 |
| Proteinase K, recombinant, Solution | FUJIFILM Wako Pure Chemical Corporation | Cat# 169-28702 |
| Glycogen Solution | nacalai tesque | Cat# 17110-11 |
| Ethanol (99.5) | nacalai tesque | Cat# 14712-63 |
| QIAquick PCR Purification Kit (250) | QIAGEN | Cat# 28106 |
| Guanidinium Chloride | nacalai tesque | Cat# 17318-95 |
| Myelin Basic Protein bovine | Sigma-Aldrich | Cat# M1891 |
| GEDTA(EGTA) | FUJIFILM Wako Pure Chemical Corporation | Cat# 342-01314 |
| (+/−)-Dithiothreitol | FUJIFILM Wako Pure Chemical Corporation | Cat# 042-29222 |
| Trichloroacetic Acid Solution (100 w/v%) | nacalai tesque | Cat# 34637-85 |
| Recombinant DNase I (RNase-free) | TaKaRa bio | Cat# 2270A |
| MiSeq Reagent Kit v3 (150-cycle) | Illumina | MS-102-3001 |
| NEBNext Ultra II DNA Library Prep Kit for Illumina | NEW ENGLAND BioLabs | E7645L |
| NEBNext Multiplex Oligos for Illumina (Index Primers Set 1) | NEW ENGLAND BioLabs | E7335S |
| **Software and algorithms** | | |
| Bowtie-1.0.0 | Langmead et al (2009) | https://genomebiology.biomedcentral.com/articles/10.1186/gb-2009-10-3-r25 |
| Samtools | Li and Durbin (2009) | https://academic.oup.com/bioinformatics/article/25/14/1754/225615 |
| MACS2 | Wilbanks and Facciotti (2010) | https://journals.plos.org/plosone/article?id=10.1371/journal.pone.0011471 |
| MEME suite | Bailey et al (2015) | https://meme-suite.org/meme/ |
| MiSeq System | Illumina | Cat# SY-410-1003 |
| S220 Focused-ultrasonicator | Covaris | Cat# 500217 |

## Cell fractionation and immunofluorescence analyses

$5.0 \times 10^7$ exponentially growing yeast cells were collected, and cell components were fractionated as previously reported (Kanoh et al, 2015). Briefly, the cell walls were digested with 100 U/ml zymolyase (Nacalai Tesque) in 1.2 M sorbitol/potassium phosphate (pH 7.0) containing 1 mM PMSF at 30°C for 30 min. The spheroplast cells, washed with 1 M sorbitol, were permeabilized in a solution containing 0.1% Triton X-100 (Sigma-Aldrich), 1.2 M sorbitol/potassium phosphate (pH 7.0) and 1 mM PMSF on ice. The cells were suspended in CSK buffer (50 mM HEPES-KOH [pH 7.5], 0.5% Triton X-100, 50 mM potassium acetate, 1 mM $MgCl_2$, 1 mM EDTA, 1 mM EGTA, 1 mM DTT, 1 mM PMSF, 0.5 mM sodium orthovanadate, 50 mM NaF, 1× protease-inhibitor cocktail (Sigma-Aldrich), 1× protease-inhibitor cocktail (Roche), and 0.1 mM MG-132) for 30 min on ice. Genomic DNA was digested with 0.25 U/ml DNase I in CSK buffer containing 10 mM $MgCl_2$ and 10 mM $CaCl_2$ and incubated at 20°C for 30 min. The cells were fixed with 4% paraformaldehyde/PBS after washing with CSK buffer. Nup98, a marker of the nuclear membrane, was detected with rat anti-Nup98 monoclonal antibody (1:500; Bioacademia) for 12 h at 4°C after blocking in PBS containing 3% BSA and 0.1% Tween 20. The cells were washed with PBS containing 0.1% Tween 20 three times, and were incubated with Alexa Fluor 488–conjugated rabbit anti–rat IgG (1:1,000; Invitrogen) in PBS containing 0.1% Tween 20 for 12 h at 4°C. Antibodies were diluted in 1% BSA in PBS and 0.1% Tween 20. Finally, the cells were stained with 1 μg/ml Hoechst 33342 for 1 h at r.t. and washed with PBS containing 0.5% Tween 20 three times before visualization under a microscope.

## Time-lapse analyses of cellular dynamics of Rif1

Cells expressing Rif1–mKO2 (red) and Taz1–EGFP (green) at the endogenous loci were analyzed under a spinning disk microscope. Images were taken as previously reported (Ito et al, 2019) with slight modification. Briefly, microscope images were acquired using an

iXon3 897 EMCCD camera (Andor) connected to Yokokawa CSU-W1 spinning-disc scan head (Yokokawa Electric Corporation) and an OlympusIX83 microscope (Olympus) with a UPlanSApo 100× NA 1.4 objective lens (Olympus) with laser illumination at 488 nm for GFP and 561 nm for mKO2. Images were captured and analyzed using MetaMorph Software (Molecular Devices). Optical section data (41 focal planes with 0.2 $\mu$m spacing every 2 min) were collected for 2 h. Time-lapse images were deconvoluted using Huygens image analysis software (Scientific Volume Imaging).

### Estimation of the number of the Rif1 molecule in fission yeast cells

His$_6$–Rif1–Flag$_3$ (93–1,400 aa) protein was expressed on ver.3-4 vector at the BamHI site, and was purified by the consecutive anti-Flag column and nickel column (Uno et al, 2012). The N-terminal 93 amino acids were removed to increase the stability of the protein. The whole cell extracts prepared by the boiling method (Takeda et al, 2001) from the cells of known numbers were serially diluted and run on PAGE together with the standard protein of the known concentrations, the purified His$_6$–Rif1–Flag$_3$ protein.

## Data Availability

The reagents, oligonucleotides, plasmids, strains, and Rif1-binding sequence lists used in this study are listed in Tables 1–3, S1, and S2.

## Supplementary Information

## Acknowledgements

This paper is dedicated to Dr. Seiji Matsumoto, a dearest friend and collaborator, who passed away on 22 November 2020, after a long fight against pancreatic cancer. Seiji contributed greatly to this work, and should have been an author of this paper. We thank Kenji Moriyama for providing the purified N-terminally truncated fission yeast Rif1 protein (93–1400 aa). We also thank Rino Fukatsu and Naoko Kakusho for excellent technical assistance. We thank Dr. Justin O'Sullivan for providing us with the data on prediction of nuclear localization of Rif1 and DNA replication in fission yeast cells. We thank Prof. Takashi Toda (Hiroshima University) for helpful suggestions. Author Contributions Y Kanoh conducted plasmid and mutant strain constructions, observed mutants by a microscope, analyzed the cell cycle by FACS, and conducted ChIP-seq and informatics analyses. M Hayano also constructed plasmids and mutant strains, characterized them, and conducted immunoprecipitation and data analysis. M Hayano and S Kudo found that Rif1 overexpression induced cell death. M Ueno provided mutant strains, conducted live cell analyses, and interpreted the data. Y Kanoh and H Masai conceived and designed the experiments and Y Kanoh and H Masai wrote the manuscript. This work was supported by JSPS KAKENHI (Grant-in-Aid for Scientific Research (A) (Grant Numbers 17H01418, 20K21410 and 20H00463 (to H Masai)); Grant-in-Aid for Scientific Research on Innovative Areas (Grant Numbers 19H05277, 20H05399 and 21H00264 (to H Masai)); Fund for the Promotion of Joint International Research (Grant Number 20KK0157); Specific Research Grants from Takeda Science Foundation (to H Masai).

## Author Contributions

Y Kanoh: conceptualization, resources, data curation, formal analysis, funding acquisition, validation, investigation, visualization, methodology, writing—original draft, and project administration.
M Ueno: resources, investigation, and methodology.
M Hayano: resources, investigation, and data curation.
S Kudo: resources and investigation.
H Masai: conceptualization, data curation, supervision, funding acquisition, investigation, visualization, methodology, project administration, and writing—original draft, review, and editing.

## Conflict of Interest Statement

The authors declare that they have no conflict of interest.

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
