## [Reviewer comments · Life Science Alliance]

Life Science Alliance

Aberrant association of chromatin with nuclear periphery induced by Rif1 leads to mitotic defect

Yutaka Kanoh, Masaru Ueno, Motoshi Hayano, Satomi Kudo, and Hisao Masai

DOI: <https://doi.org/10.26508/lsa.202201603>

Corresponding author(s): Hisao Masai, Tokyo Metropolitan Institute of Medical Science

Review Timeline:

Submission Date:	2022-07-13
Editorial Decision:	2022-09-02
Revision Received:	2022-12-03
Editorial Decision:	2023-01-13
Revision Received:	2023-01-23
Accepted:	2023-01-24

Transaction Report:

September 2, 2022

Re: Life Science Alliance manuscript #LSA-2022-01603-T

Dr. Hisao Masai

-

Department of Cell Biology
Tokyo Metropolitan Institute of Medical Science
Tokyo 113-8613

Dear Dr. Masai,

Thank you for submitting your manuscript entitled "Aberrant association of chromatin with nuclear periphery induced by Rif1 leads to mitotic defect" to Life Science Alliance. The manuscript was assessed by expert reviewers, whose comments are appended to this letter. We invite you to submit a revised manuscript addressing the Reviewer comments.

Thank you for this interesting contribution to Life Science Alliance. We are looking forward to receiving your revised manuscript.

Sincerely,

B. MANUSCRIPT ORGANIZATION AND FORMATTING:

Reviewer #1 (Comments to the Authors (Required)):

The manuscript describes how Rif1 overexpression causes growth inhibition in fission yeast. The findings are potentially interesting, but the current manuscript contains serious flaws in its writing, not in scientific contents, making it difficult to read through the results. Overall, figures are not well explained or not well displayed.

It is difficult to make comments without page or line numbers, so I paginated the manuscript from the title page (page 1) to the end of References (page 34) with an empty page of page 5. I refer these page numbers in the following specific comments.

1. "We measured the numbers of Rif1 molecules --- (Fig EV1B)" on page 6: It is not explained how these numbers were estimated. Legend to Figure EV1B contains too many mistakes and inconsistency: check strain names, with or without thiamine, His6-Rif1-Flag3 or His6-spRif1-Flag3. Make the description consistent in the text, legend, and figure.
2. Bottom lines in page 6: Results in Fig 1C shows that the OP toxic domain resides somewhere in 81-150aa. It seems difficult to attribute the toxicity to HEAT repeats in the N-terminal domain 88-1023aa.
3. The text refers KVxF and GILK motifs (page 7), but Figure 2B only shows RVxF and SILK motifs.
4. The text says "DNA synthesis at 18.5 hr from the release" and so on (page 8). This is not the time after release. The time should be 30 min after release because cells were released from the arrest at 18 hr after start of culture. Descriptions of time course must be consistent in Figure 3A legend, Methods, and the text.
5. Figure 3C and 3D are not cited in the text (page 8). It is not obvious which figures demonstrate "While Cds1 kinase activity decreased at 6 hr after induction ---". Describe more details in this paragraph with citation of Figure 3.
6. On page 9, "Cut cells appeared in Rif1-overexpressing cells - (Fig 4A)". First, the "cut" phenotype is not obvious for general readers and must be defined. Fig 4A is a bright-field image, and the "cut" phenotype cannot be seen in bright-field images. The arrowhead in Fig 4A doesn't look like the "cut" phenotype although the figure legend claims so. Defining the "cut" phenotypes requires fluorescence images of chromatin, such as histone-GFP in Fig 4C. Define the "cut" and other phenotypes in Fig 4C, and present those fractions in Fig 4B. Fig 4A could be omitted.
7. SAC should be spelled out at the first appearance (page 9). In the same paragraph, the text says "in these mutants (Fig. 5B)", but no mutants are mentioned. They should be called "mad2 Δ and bub1 Δ " here.
8. In the top paragraph in page 10 (Figure 5C), define the time 0.
9. Figure 5E is not well explained in the text (page 10) or in the legend. The "cut" phenotype cannot be seen in the bright field images in Figure 5E.
10. Last complete paragraph in page 11 "This population reached over 6% with the wild type and 11 % with the PP1bs mutant -- the PP1bs mutant may be consistent with more severe growth inhibition with this mutant.": Which one of the PP1bs mutants does "the PP1bs mutant" mean? L848S or R236H mutant?
11. Nomenclatures in the text and Figure EV5B are not correlated, and thus confusing (page 12).
12. How is the localization of Rif1-mKO2 compared with that of Rif1?
13. The species of fission yeast is not specified in the main text. Although "Schizosaccharomyces pombe" first appears in Materials and Methods, this species is never linked to the fission yeast throughout the manuscript.
14. Figure 5A, label: "short < 3 μ M" should be "3 μ m".
15. No labels for each of the Supplementary movies. Relationships with Fig EV5D should be provided.

Reviewer #2 (Comments to the Authors (Required)):

The manuscript titled "Aberrant association with nuclear periphery induced by Rif1 leads to mitotic defect" by Kanoh et al. is well written. The experimental data were properly analyzed and clearly presented. The conclusion, based on the available data, is appropriate. The finding described in this manuscript should be useful for the future study of Rif1 and its biological functions. Thus, I suggest the publication of this manuscript with minor revisions.

1. In Fig. 6B, the authors did ChIP-seq against Rif1 in the Rif1-overexpressed cells. It is interesting to know what other sequences Rif1 binds to when the number of Rif1 molecules jumps from 1000 per cell to 10,000-37,000 per cell besides Rif1 binding to G4 structures. Please list those sequences in a table.

2. Besides binding to DNA, what proteins are associated with Rif1? Please have a paragraph to discuss this matter in the section of Discussion.

Reviewer #3 (Comments to the Authors (Required)):

Kanoh et al. report the phenotype of Rif1 overexpression in fission yeast. They find that such over expression is lethal. The terminal phenotype includes missegregated DNA, metaphase spindle accumulation and redistribution of the chromatin to the nuclear periphery. They further show that these phenotypes do not require Rif1's well-studied interaction with PP1, but that over expression of Rif1 that does bind PP1 also inhibits replication initiation. The work appears to be well done and the results robust, although attention to the following suggestions would strengthen the work. However, the authors conclusions are overly strong. In particular, their claims in the Abstract that "regulated Rif1-mediated chromatin association with nuclear periphery is important for coordinated progression of S- and M-phases" and in the Introduction that "association of chromatin with nuclear periphery needs to be precisely regulated for proper S- and M-phase progression" and speculative and not directly supported by their data. More generally, a major weakness of the paper is that the Rif1 overexpression terminal phenotype is quite heterogeneous and pleiotropic, with many of the reported phenotypes showing low penetrance. Therefore, without a significantly more detailed analysis, it is impossible to say which, if any, of the observed phenotypes are the primary cause of cell death and which are secondary consequences. It is, for example, possible that the primary cause of cell death is disruption of transcription and that the observed phenotypes are a consequence of gene misexpression, a possibility the authors should discuss.

In Figure 2, the authors show that overexpression of PP1-binding-defective Rif1 arrests growth. An important control is to show that it does not delay replication. They show as much in Figure 3, but they should indicate this fact when discussing the results in Figure 2.

It is strange that deleting rif1 partially restores growth to cells overexpressing DDK. Rif1 is thought to antagonize DDK, so the opposite result is expected. The authors should discuss this point.

The fact that the growth inhibition caused by Rif1 overexpression is not affected by *cdc25-22* or *wee1-50* is not informative. Although both genes are involved in the G2/M (not mitotic) checkpoints, neither allele has strong checkpoint defects.

Figures 1A, 2D and 3B should be quantitated and the quantitation should be reported below each lane. Quantitation relative to wild type would suffice, but absolute quantitation, based on Figure EV1, would be even better. Also, the number of repeats should be reported in the figure legend. If the experiment has not been repeated, it should be.

Figure 1 C should be grayscale, like the other panels.

Figure 3C: The band should be labeled

Figure 3D: Since the X axis is a continuous variable (time), this panel should be a line graph with the time points distributed proportionally and lines connecting (interpolating) the data points at each time. Also, since the Y axis units are relative, they should go from 0 to 6, not 0 to 6,000.

Figure 9: "Overproduction" is misspelled.

The authors should indicate in the Results, Methods and Figure Legends the stain names (as listed in the Stain List table) of the strains used in each experiment.

All the comments from the reviewers are in black and our responses are in blue.

Reviewer #1 (Comments to the Authors (Required)):

The manuscript describes how Rif1 overexpression causes growth inhibition in fission yeast. The findings are potentially interesting, but the current manuscript contains serious flaws in its writing, not in scientific contents, making it difficult to read through the results. Overall, figures are not well explained or not well displayed.

It is difficult to make comments without page or line numbers, so I paginated the manuscript from the title page (page 1) to the end of References (page 34) with an empty page of page 5. I refer these page numbers in the following specific comments.

We apologize for not paginating the manuscript and not presenting the line numbers. We now added the page and line numbers to the revised manuscript.

1. "We measured the numbers of Rif1 molecules --- (Fig EV1B)" on page 6: It is not explained how these numbers were estimated. Legend to Figure EV1B contains too many mistakes and inconsistency: check strain names, with or without thiamine, His6-Rif1-Flag3 or His6-spRif1-Flag3. Make the description consistent in the text, legend, and figure.

We apologize for the errors and inconsistency.

The endogenous Rif1-Flag₃ is expressed at the endogenous locus under its own promoter. Therefore, its expression is not affected by the presence or absence of thiamine. This serves as the control level for the endogenous Rif1 protein. The strain carrying the plasmid pREP41-Rif1-Flag₃ expresses Rif1-Flag₃ under nmt1 promoter. Therefore, the Rif1-Flag₃ protein is overexpressed when grown in the absence of thiamine.

The control Rif1 protein, His₃-Rif1(93-1400aa)-Flag₃, used as a standard for quantification of the cellular Rif1 protein, was overproduced and purified from mammalian cells. Although the protein is not the same as the Rif1-Flag₃ expressed in yeast cells, and contains a small N-terminal truncation to prevent degradation during purification, it serves well as a standard.

We have checked the figures and text carefully and made sure there is no inconsistency. We also added more explanation on how the numbers were estimated.

2. Bottom lines in page 6: Results in Fig 1C shows that the OP toxic domain resides somewhere in 81-150aa. It seems difficult to attribute the toxicity to HEAT repeats in the N-terminal domain 88-1023aa.

As the reviewer points out, the segment 81-150aa is required for growth inhibition by overexpression. As shown in Figure 1D, the segment 1-1260 is sufficient for inhibition,

suggesting that 81-1260 is sufficient for inhibition, although this was not experimentally examined. The segment 1-965 does not show toxicity, thus suggesting that 966-1260 is important.

As we stated in the text, 88-1023 is predicted to form HEAT repeats, and therefore, the presence of the intact HEAT/Armadillo repeat structure may be required and sufficient for the growth inhibition. Since we have not tested if 88-1023 can cause growth inhibition, we cannot conclude that the HEAT domain alone is sufficient for growth inhibition. We have modified the text so that we do not overstate on our data.

3. The text refers KVxF and GILK motifs (page 7), but Figure 2B only shows RVxF and SILK motifs.

The PP1 binding motif of *S. pombe* Rif1 is somewhat diverged from the consensus RVxF-SILK. We have used SILK motif in the revised manuscript.

4. The text says "DNA synthesis at 18.5 hr from the release" and so on (page 8). This is not the time after release. The time should be 30 min after release because cells were released from the arrest at 18 hr after start of culture. Descriptions of time course must be consistent in Figure 3A legend, Methods, and the text.

The reviewer is correct. 18.5 hr is 30 min after the release. We have corrected this in the text.

5. Figure 3C and 3D are not cited in the text (page 8). It is not obvious which figures demonstrate "While Cds1 kinase activity decreased at 6 hr after induction ---". Describe more details in this paragraph with citation of Figure 3.

We apologize for the overlook. Figure 3C and D are now cited. "The decrease of Cds1 kinase activity at 6 hr" was based on the data on Figure 3C, lane 3. "6 hr" should have been "12 hr" after release, and this was also corrected.

6. On page 9, "Cut cells appeared in Rif1-overexpressing cells - (Fig 4A)". First, the "cut" phenotype is not obvious for general readers and must be defined. Fig 4A is a bright-field image, and the "cut" phenotype cannot be seen in bright-field images. The arrowhead in Fig 4A doesn't look like the "cut" phenotype although the figure legend claims so. Defining the "cut" phenotypes requires fluorescence images of chromatin,

such as histone-GFP in Fig 4C. Define the "cut" and other phenotypes in Fig 4C, and present those fractions in Fig 4B. Fig 4A could be omitted.

After analyzing the data, we agree with the reviewer that cut cells need to be more precisely defined. We decided to categorize the cells into "unequally segregated" and "entangled" (cut cells were included in "unequally segregated")(new Fig. 4A). In Figure 5E, we scored cells with aberrant morphology in stead of "cut" cells. We have omitted the old Figure 4A.

7. SAC should be spelled out at the first appearance (page 9). In the same paragraph, the text says "in these mutants (Fig. 5B)", but no mutants are mentioned. They should be called "mad2 Δ and bub1 Δ " here.

We have made these corrections.

8. In the top paragraph in page 10 (Figure 5C), define the time 0.

Time 0 is the timing when Sad1 foci divide. We made this clear in the revised manuscript (in the text and in the legend for Figure 5E).

9. Figure 5E is not well explained in the text (page 10) or in the legend. The "cut" phenotype cannot be seen in the bright field images in Figure 5E.

10. Last complete paragraph in page 11 "This population reached over 6% with the wild type and 11 % with the PP1bs mutant --- the PP1bs mutant may be consistent with more severe growth inhibition with this mutant.": Which one of the PP1bs mutants does "the PP1bs mutant" mean? L848S or R236H mutant?

We apologize that we were not clear enough for our description. "the PP1bs mutant" is the one described in Figure 2B, and is expressed only on a plasmid. L848S and R236H mutants are *rif1* mutants we have isolated and are not PP1bs mutants. The former is incapable of chromatin binding while the latter can bind to chromatin.

We made this clear in the text.

11. Nomenclatures in the text and Figure EV5B are not correlated, and thus confusing (page 12).

We have unified the nomenclature.

Rif1:mKO2 -> Rif1-mKO2

12. How is the localization of Rif1-mKO2 compared with that of Rif1?

Similar to Rif1-mKO2, the major endogenous Rif1 signals appear at telomeres, colocalizing with Taz1-GFP, in addition to dispersed non-telomeric nuclear signals. Colocalization of Rif1 and telomere was previously reported by using Taz1-HA, Rap1-HA Rif1-Myc cells (Ref. Kanoh J, Ishikawa F 2001).

13. The species of fission yeast is not specified in the main text. Although "Schizosaccharomyces pombe" first appears in Materials and Methods, this species is never linked to the fission yeast throughout the manuscript.

We stated "Schizosaccharomyces pombe" at the first appearance of fission yeast in the Abstract and in Introduction

14. Figure 5A, label: "short < 3μM" should be "3μm".

This was corrected.

15. No labels for each of the Supplementary movies. Relationships with Fig EV5D should be provided.

The movies are now labeled with titles, as follows.

Figure EV5D contains snapshots from the movies.

Movie1_Rif1mKO2_related_with_FigEV5D

Movie2_Taz1GFP_related_with_FigEV5D

Movie3_Merged_Rif1_Taz1_related_with_FigEV5D

Movie4_3D_Merged_Rif1_Taz1_related_with_FigEV5D

Movie5_Trimmed_Movie3_related_with_FigEV5D

Reviewer #2 (Comments to the Authors (Required)):

The manuscript titled "Aberrant association with nuclear periphery induced by Rif1 leads to mitotic defect" by Kanoh et al. is well written. The experimental data were properly analyzed and clearly presented. The conclusion, based on the available data, is appropriate. The finding described in this manuscript should be useful for the future study of Rif1 and its biological functions. Thus, I suggest the publication of this manuscript with minor revisions.

1. In Fig. 6B, the authors did ChIP-seq against Rif1 in the Rif1-overexpressed cells. It is interesting to know what other sequences Rif1 binds to when the number of Rif1 molecules jumps from 1000 per cell to 10,000-37,000 per cell besides Rif1 binding to G4 structures. Please list those sequences in a table.

Peak lists for the wild-type and Rif1 overexpression strain are presented and the 300 bp sequences of each peak have been extracted and presented in the excel file (Tables 4 and 5).

2. Besides binding to DNA, what proteins are associated with Rif1? Please have a paragraph to discuss this matter in the section of Discussion.

The following was added to Discussion

“In addition to conserved interaction with PP1, Rif1 is known to interact with a number of proteins. *S.pombe* Rif1 interacts with telomere factors Taz1 and Rap1. This interaction is important for its function at the telomere. It also interacts with Epe1, Jmjc domain chromatin associated protein, suggesting its potential role in chromatin regulation. Human Rif1 interacts with DSB repair factors, 53BP1, Mdc1, Bloom RecQ helicase, and anti-silencing function 1B histone chaperone, ASF1B. This underscores its roles in regulation of DSB repair and epigenomic state.”

Reviewer #3 (Comments to the Authors (Required)):

Kanoh et al. report the phenotype of Rif1 overexpression in fission yeast. They find that such over expression is lethal. The terminal phenotype includes miss-segregated DNA, metaphase spindle accumulation and redistribution of the chromatin to the nuclear periphery. They further show that these phenotypes do not require Rif1's

well-studied interaction with PP1, but that over expression of Rif1 that does bind PP1 also inhibits replication initiation. The work appears to be well done and the results robust, although attention to the following suggestions would strengthen the work. However, the authors conclusions are overly strong. In particular, their claims in the Abstract that "regulated Rif1-mediated chromatin association with nuclear periphery is important for coordinated progression of S- and M-phases" and in the Introduction that "association of chromatin with nuclearperiphery needs to be precisely regulated for proper S- and M-phase progression" and speculative and not directly supported by their data. More generally, a major weakness of the paper is that the Rif1 overexpression terminal phenotype is quite heterogeneous and pleiotropic, with many of the reported phenotypes showing low penetrance. Therefore, without a significantly more detailed analysis, it is impossible to say which, if any, of the observed phenotypes are the primary cause of cell death and which are secondary consequences. It is, for example, possible that the primary cause of cell death is disruption of transcription and that the observed phenotypes are a consequence of gene misexpression, a possibility the authors should discuss.

We thank this reviewer for very important comments. We agree that our conclusion that "regulated Rif1-mediated chromatin association with nuclear periphery is important for coordinated progression of S- and M-phases" may be too strong. We have modified the abstract and relevant text to weaken our statement. Specifically, we have incorporated the possibilities indicated by this reviewer and other possibilities at the end of Discussion (see below), and weakened our conclusions.

“However, we cannot rule out the possibility that the phenotypes we observe upon Rif1 overexpression could be secondary consequences of its effect on transcription or on other chromosomal events including repair and recombination. More detailed studies will be needed to precisely determine the effects of deregulated chromatin association with nuclear membrane on cell cycle progression and cell survival.”

In Figure 2, the authors show that overexpression of PP1-binding-defective Rif1 arrests growth. An important control is to show that it does not delay replication. They show as much in Figure 3, but they should indicate this fact when discussing the results in Figure 2.

We have mentioned the inability of PP1 mutant to delay DNA replication when we discuss the result of Figure 2. See below.

“As shown in the later section, the PP1bs mutant of Rif1 loses the ability to inhibit DNA synthesis, indicating that growth inhibition by Rif1 is related to the events other than S phase.”

It is strange that deleting *rif1* partially restores growth to cells overexpressing DDK. Rif1 is thought to antagonize DDK, so the opposite result is expected. The authors should discuss this point.

The overexpression of Hsk1 and Dfp1/Him1 is very toxic even to the wild-type cells. This appears to be due to inhibition of S phase progression. That may be the reason why *rif1*Δ shows some recovery of the DDK overproducing cells.

The fact that the growth inhibition caused by Rif1 overexpression is not affected by *cdc25-22* or *wee1-50* is not informative. Although both genes are involved in the G2/M (not mitotic) checkpoints, neither allele has strong checkpoint defects.

We would like to thank the reviewer for the important comment. In accordance with this comment, we have modified the text explaining the result of *cdc25-22* and *wee1-50* as shown below.

“The extent of growth inhibition was not affected by *cdc25-22* or *wee1-50*, genes involved in mitosis (Iino & Yamamoto, 1997; Kumar & Huberman, 2004; Rowley *et al*, 1992) (**Fig EV2C**). These results suggest that the growth inhibition is not caused by replication or mitotic checkpoint functions or deregulation of mitotic kinases.”

Figures 1A, 2D and 3B should be quantitated and the quantitation should be reported below each lane. Quantitation relative to wild type would suffice, but absolute quantitation, based on Figure EV1, would be even better. Also, the number of repeats should be reported in the figure legend. If the experiment has not been repeated, it should be.

We have repeated the experiment for Fig1A,2D,3B multiple times and quantified the data. The data are now shown in the graphs.

Figure 1 C should be grayscale, like the other panels.

We have changed to gray scale.

Figure 3C: The band should be labeled

Cds1 protein band is indicated in Fig 3C.

Figure 3D: Since the X axis is a continuous variable (time), this panel should be a line graph with the time points distributed proportionally and lines connecting (interpolating) the data points at each time. Also, since the Y axis units are relative, they should go from 0 to 6, not 0 to 6,000.

In accordance with this comment, we changed the graph format as suggested. The Y axis unit was also changed.

Figure 9: "Overproduction" is misspelled.

Corrected

The authors should indicate in the Results, Methods and Figure Legends the stain names (as listed in the Stain List table) of the strains used in each experiment.

We have made the list of all the strains used in the manuscript, and indicated in which figures each strain was utilized.

I hope these revision and explanation answer all the comments by the reviewers and the revised manuscript is now acceptable for publication in Life Science Alliance

Sincerely,

Hisao Masai

Genome Dynamics Project,

Department of Basic Medical Sciences

Tokyo Metropolitan Institute of Medical Science,

2-1-6 Kamikitazawa, Setagaya-ku
Tokyo 156-8506, JAPAN
Tel: +81-3-5316-3231
Fax: +81-3-5316-3145
E-mail: masai-hs@igakuken.or.jp

January 13, 2023

RE: Life Science Alliance Manuscript #LSA-2022-01603-TR

Dr. Hisao Masai
Tokyo Metropolitan Institute of Medical Science
Department of Cell Biology
Tokyo Metropolitan Institute of Medical Science
Tokyo 113-8613
Japan

Dear Dr. Masai,

Thank you for submitting your revised manuscript entitled "Aberrant association of chromatin with nuclear periphery induced by Rif1 leads to mitotic defect". We would be happy to publish your paper in Life Science Alliance pending final revisions necessary to meet our formatting guidelines.

- please consult our manuscript preparation guidelines <https://www.life-science-alliance.org/manuscript-prep> and make sure your manuscript sections are in the correct order
- please add the Twitter handle of your host institute/organization as well as your own or/and one of the authors in our system
- please remove any authors in your manuscript that are not listed in our system
- please add the Author contributions and a conflict of interest statement to the main manuscript text
- please rename your EV figures as supplementary figures and adjust your figure callouts in the text accordingly

Figure Check:

- Figure 5E, Figure EV6 C: scale bars needed

A. FINAL FILES:

B. MANUSCRIPT ORGANIZATION AND FORMATTING:

Sincerely,

Reviewer #1 (Comments to the Authors (Required)):

The manuscript describes how Rif1 overexpression causes growth inhibition in fission yeast.

The initial manuscript contained several problems and errors. Those problems have been properly addressed during the revision. I have no further concerns.

Reviewer #2 (Comments to the Authors (Required)):

I am satisfied with the revision. Therefore, I support the publication of this manuscript.

January 24, 2023

RE: Life Science Alliance Manuscript #LSA-2022-01603-TRR

Dr. Hisao Masai
Tokyo Metropolitan Institute of Medical Science
Basic Medical Sciences
2-1-6 Kamikitazawa
Setagaya-ku
Tokyo 156-8506
Japan

Dear Dr. Masai,

Thank you for submitting your Research Article entitled "Aberrant association of chromatin with nuclear periphery induced by Rif1 leads to mitotic defect". It is a pleasure to let you know that your manuscript is now accepted for publication in Life Science Alliance. Congratulations on this interesting work.

DISTRIBUTION OF MATERIALS:

Again, congratulations on a very nice paper. I hope you found the review process to be constructive and are pleased with how the manuscript was handled editorially. We look forward to future exciting submissions from your lab.

Sincerely,
